# TcrXY is an acid-sensing two-component transcriptional regulator of *Mycobacterium tuberculosis* required for persistent infection

Miljan Stupar ®[1], Lendl Tan[1], Edward D. Kerr ®[1], Christopher J. De Voss ®[1], Brian M. Forde ®[2], Benjamin L. Schulz ®[1] & Nicholas P. West ®[1] ✉

The ability of *Mycobacterium tuberculosis* (Mtb) to persist in the host complicates and prolongs tuberculosis (TB) patient chemotherapy. Here we demonstrate that a neglected two-component system (TCS) of Mtb, TcrXY, is an autoregulated acid-sensing TCS that controls a functionally diverse 70-gene regulon required for bacterial persistence. Characterisation of two representatives of this regulon, *Rv3706c* and *Rv3705A*, implicate these genes as key determinants for the survival of Mtb in vivo by serving as important effectors to mitigate redox stress at acidic pH. We show that genetic silencing of the response regulator *tcrX* using CRISPR interference attenuates the persistence of Mtb during chronic mouse infection and improves treatment with the two front-line anti-TB drugs, rifampicin and isoniazid. We propose that targeting TcrXY signal transduction blocks the ability of Mtb to sense and respond to acid stress, resulting in a disordered program of persistence to render the organism vulnerable to existing TB chemotherapy.

Tuberculosis (TB) is an ongoing pandemic with enormous implications for human health and productivity. Despite the availability of effective antibiotics, successful treatment remains disappointing with cure rates of 85% for drug-susceptible TB and just 60% for drug-resistant TB[1]. Protracted treatment regimens for TB are necessary due to the capacity of *Mycobacterium tuberculosis* (Mtb), the bacteria responsible, to enter and remain in a state of in vivo persistence during host selective pressures. Current strategies are failing to address the innate phenotypic resistance displayed by Mtb, and this critical aspect remains a significant impediment to efficient therapy[1,2].

As a chronic intracellular pathogen, Mtb is reliant on the faithful assimilation of survival signals for persistence[3]. Continuous surveillance of the host environment, together with rapid translation of the complex array of stimuli into appropriate transcriptional responses, is required for the infection–persistence–transmission cycle. The two-component system (TCS) of transcriptional regulation is most appropriately assigned this function and is vital in this role. Mtb contains 12 complete TCSs operating singularly or in concert for efficient

signal transduction[4], often resulting in the onset of persistence and drug tolerance. Innate and immunological challenges, including hypoxia[5,6], nitric oxide[7], the limitation of nutrients[8,9], and the acidified environment of the macrophage phagosome[10–12] represent significant environmental cues, with TCSs responsible for the initial responses required for adaptation to many of these stimuli.

The immediate reduction of intraphagosomal pH following phagocytosis is a central macrophage-directed strategy for effective microbial defence[13], however Mtb, through phenotypic acid tolerance[12,14–16], metabolic remodelling[10,17] and manipulation of its phagosome[18,19], can evade clearance. While the critical nature of pH-induced gene regulation and phenotypic modification of Mtb is not in dispute, the bacterial mechanisms responsible for the detection of this key defence signal are not well understood. We reasoned that further TCS input would most likely support the essential rapid responses required in an acidifying intracellular environment. Here we report the characterisation of the neglected TCS, TcrX/TcrY (Rv3765c/Rv3764c), as a pH-sensing signal transduction pathway required for detecting and responding to intramacrophage pH stress

---

[1]School of Chemistry and Molecular Biosciences, Australian Infectious Disease Research Centre, The University of Queensland, Brisbane, Australia. [2]Faculty of Medicine, UQ Centre for Clinical Research, The University of Queensland, Brisbane, Australia. ✉e-mail: n.west@uq.edu.au

for persistence. Our findings indicate that a strategy to confound the bacterial detection of host responses can render Mtb susceptible to immunological and/or antimicrobial clearance to potentially shorten therapy duration.

## Results

### TcrXY is an autoregulated acid-responsive TCS in Mtb

The Mtb transcriptional response to acidic pH is expansive and results in significant remodelling of Mtb physiology. We focused our efforts on one poorly characterised pH-inducible operon[11,20], *tcrXY*, encoding a bona fide TCS comprised of the sensor histidine kinase TcrY and cognate response regulator, TcrX[21]. In our hands, transcription of *tcrX* and *tcrY* were induced ~5-fold and ~2-fold, respectively, in Mtb H37Rv cultured in acidified minimal growth media (pH 5.4) for 24 h, compared to neutral growth media (pH 7) (Supplementary Fig. 1a). This was consistent in a nutrient-rich growth media (7H9 complete, pH 5.4) (Supplementary Fig. 1b). The kinetics of *tcrX* mRNA levels displayed a robust activation surge reaching a maximum after 24 h, that diminished by 72 h (Supplementary Fig. 1c). These data confirm prior reports that *tcrXY* is an acid-inducible locus in Mtb[11,20].

To better study transcriptional activation of the *tcrXY* operon, we developed a Mtb dual transcriptional reporter, where the transcription of *tcrXY* is linked to mCherry expression, in a constitutive GFP expressing background to serve as an internal control. The resulting reporter strain, H37Rv (P$_{tcrXY}$::*mCherry*; P$_{rpoB}$::*gfp*), displayed pH- and time-dependent activation (~2-fold after 72 h) (Fig. 1a). A one pH-unit decrease in culture medium acidity (pH 6) was sufficient to induce the reporter strain from baseline (pH 7); the pH corresponding to the Mtb-containing phagosome in resting macrophages[22] (Fig. 1b). Maximum activity was observed at pH 5.4, the pH corresponding to matured phagosomes of activated macrophages[22,23]. Reporter activity was diminished in highly acidic culture media (pH < 5), albeit remained above baseline levels (pH 7). This indicates that *tcrXY* activity levels closely resemble the maturation status of the Mtb-containing phagosome. Next, we confirmed whether pH-dependent activation of *tcrXY* in culture translated into similar responses during infection of macrophages. Murine BMDMs were infected with H37Rv (P$_{tcrXY}$::*mCherry*; P$_{rpoB}$::*gfp*) for 24 h and acidic compartments stained using LysoTracker (Fig. 1d). A statistically significant 1.5X increase in median mCherry signal was observed in bacteria that co-localised with LysoTracker, as compared to bacteria in non-acidic compartments (Fig. 1e). Immunological activation of BMDMs prior to infection increases the acidification of the Mtb phagosome[22,23], and indeed this translated into greater activation of *tcrXY* transcription in LysoTracker-positive compartments (2X increase in median mCherry signal), suggesting a causal, intracellular link. Collectively, these data support *tcrXY* as an intramacrophage pH-sensitive locus.

Since many TCSs are positively autoregulated in response to their inducing stimulus[24], we wondered whether TcrXY autoregulation was responsible for *tcrXY* operon induction at acidic pH. We constructed a Δ*tcrXY* mutant in H37Rv, and confirmed its genetic integrity by PCR (Supplementary Fig. 2) and whole genome sequencing, verifying the absence of any additional mutations relative to the parental strain. Introduction of the dual reporter construct (P$_{tcrXY}$::*mCherry*; P$_{rpoB}$::*gfp*) into Δ*tcrXY* revealed that *tcrXY* induction at pH 5.4 was completely dependent on the TcrXY TCS (Fig. 1f). To further explore whether the loss of *tcrXY* expression was specifically due to a loss of TcrXY signalling at acidic pH, we introduced into Δ*tcrXY* a modified *tcrXY* allele where the TcrX or TcrY phosphorylation site was genetically substituted to a non-phosphorylatable amino acid (D54A and H256Q, respectively). Induction of *tcrX* mRNA at acidic pH was hindered in both phosphorylation mutants to a different yet significant magnitude (~2-fold and ~0.75-fold, respectively), as compared to a wild-type *tcrXY* allele (~4-fold) (Fig. 1g). These results suggest that the system is clearly

intricate and not fully understood, with the possibility of other factors affecting their expression.

As TcrX is a transcriptional regulator, we assessed whether the autoregulation of *tcrXY* was mediated specifically by its DNA-binding activity. We optimised a protocol to purify TcrX under non-denaturing conditions (Supplementary Fig. 3) and assessed its activity using an electrophoretic mobility shift assay. We observed a direct interaction between TcrX and P$_{tcrXY}$ (Supplementary Fig. 4a, b), with its affinity for P$_{tcrXY}$ being enhanced by phosphorylation of TcrX (P-TcrX) (Supplementary Fig. 4a, b). A phosphorylation-deficient protein, TcrX$^{D54A}$, was able to bind to P$_{tcrXY}$, as previously reported[25], however it was unaffected when attempts were made to phosphorylate it (Supplementary Fig. 4b). We assessed the specificity of the P-TcrX-P$_{tcrXY}$ interaction using unlabelled specific and non-specific competitor DNA (Supplementary Fig. 4c). Next, we replaced the putative TcrX-binding site in the *tcrXY* promoter, a perfect 7-bp mirror repeat[25], with a scrambled nucleotide sequence and confirmed that the loss of this putative site abolished binding (Supplementary Fig. 4d). Both sequences of the mirror repeat (BS1 and BS2) were required for DNA binding. Mtb reporter strains harbouring these altered promoter sequences displayed a complete loss of pH-dependent activity (Fig. 1f). We observed very low promoter activity when we scrambled BS2 or both BS1 and BS2 and suspect this is due to the overlap of the BS2 site with the putative −10 promoter sequence (Supplementary Fig. 5). Overall, we now propose a model where TcrXY signalling is activated in response to intramacrophage acidic pH to auto-induce the *tcrXY* operon (Fig. 1h).

### The loss of TcrXY signalling attenuates Mtb persistence

We sought to investigate the importance of TcrXY acid sensing during infection. In mice, the Δ*tcrXY* mutant replicated comparably to wild-type H37Rv in the lungs during the acute phase of infection (Fig. 2). During the chronic phase of infection, Δ*tcrXY* was attenuated for survival. At 16 weeks post-aerosol infection, Δ*tcrXY* titres in the lungs, spleen and liver were reduced 10-fold compared to H37Rv (Fig. 2). Complementation with functional *tcrXY* (*tcrXY*-comp) restored the wild-type persistence phenotype. This indicates that the loss of the TcrXY TCS attenuates Mtb survival in the chronic phase of TB, highlighting the importance of TcrXY regulation for adaptation to acidic pH and thus, long-term persistence.

### The loss of TcrXY signalling has significant impacts on pH-dependent gene expression

To assess the transcriptomic impact of TcrXY signalling during acid stress, we sequenced mRNA isolated from Mtb H37Rv and Δ*tcrXY* cultured in minimal media at pH 7 or pH 5.4 for 24 h. At acidic pH, the loss of *tcrXY* had a significant impact on the Mtb transcriptome, resulting in the upregulation of 66 genes and downregulation of 4 genes (fold-change > 2; adj. *P* < 0.05) (Fig. 3a and Supplementary Data 1a, b). This indicates that TcrXY is predominantly a negative regulator of gene expression at acidic pH. In stark contrast, we observed little impact on the Mtb transcriptome at neutral pH (Supplementary Data 1c, d), highlighting the importance of acid stress as an activator of TcrXY regulation.

We observed the dysregulation of several pH-dependent genetic pathways upon disruption of TcrXY signal transduction. This included genes involved in mycolic acid remodelling, redox homoeostasis, fatty acid metabolism, as well as genes encoding low molecular weight chaperones and toxin-antitoxin systems (Fig. 3b). Overall, these data indicate that TcrXY has a significant impact on Mtb physiology at acidic pH.

### TcrXY directly regulates two novel small proline-rich proteins at acidic pH

To gain insight into the TcrX-dependent adaptations responsible for attenuated Mtb persistence, we focused our efforts on two

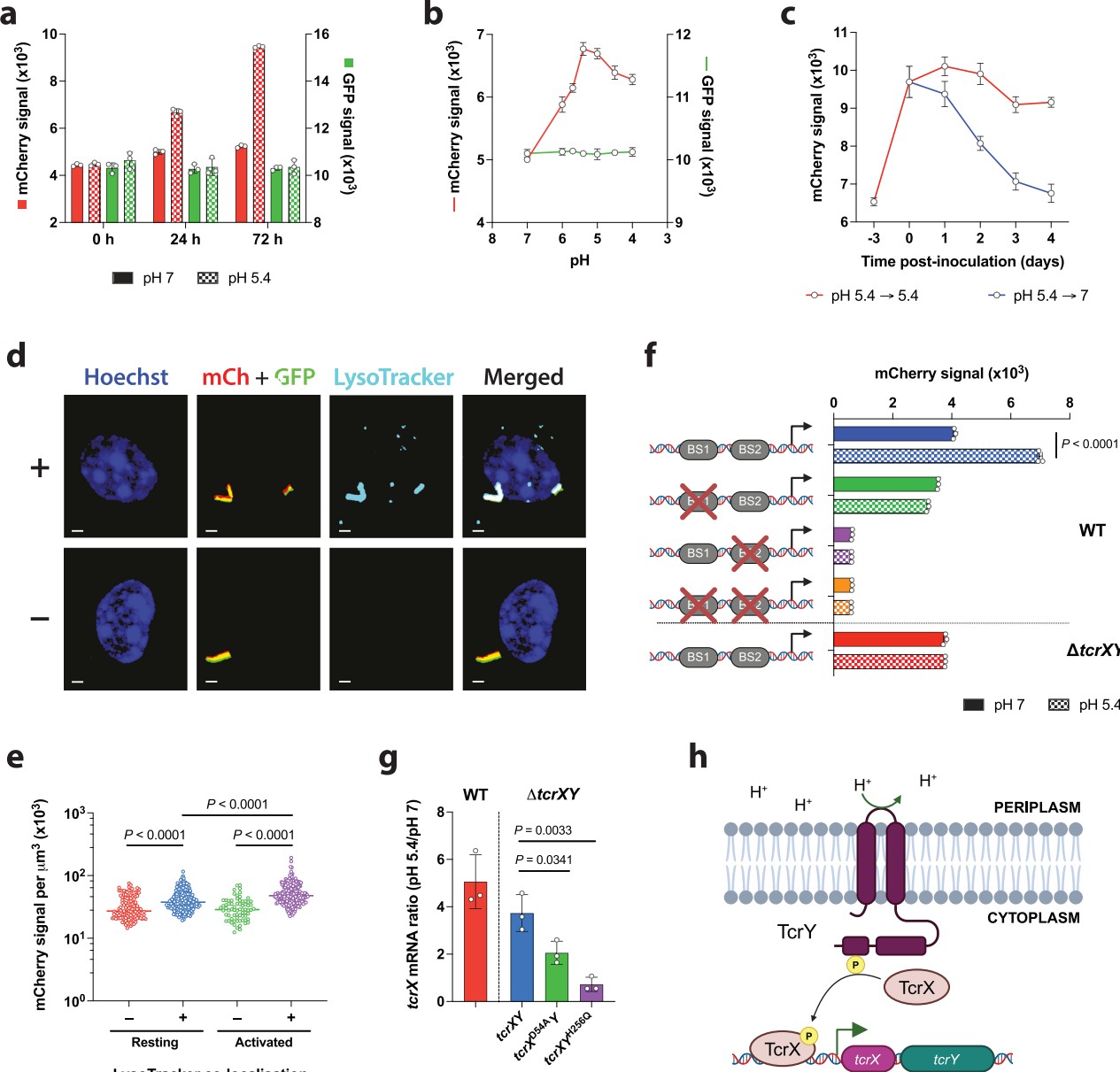

**Fig. 1 | TcrXY is an autoregulated acid-sensing TCS in Mtb. a** $P_{tcrXY}$ and $P_{rpoB}$ activity in Mtb H37Rv cultured in pH 7 or pH 5.4 media. Each bar represents the median mCherry or GFP fluorescence intensity from three biological replicates (open circle). **b** $P_{tcrXY}$ and $P_{rpoB}$ activity in Mtb H37Rv as a function of culture medium acidity after 24 h. Each point represents the median mCherry or GFP fluorescence intensity from three biological replicates. **c** $P_{tcrXY}$ activity in Mtb H37Rv following the removal of its activating stimulus. Each point represents the median mCherry fluorescence intensity from three biological replicates. **d** Exemplar confocal micrographs of resting murine BMDMs infected with the Mtb *tcrXY* reporter for 24 h and stained with LysoTracker. Images depict cell nucleus (blue, Hoechst), Mtb *tcrX* reporter strain (red, mCherry (mCh) and green, GFP) and acidic compartments (cyan, LysoTracker). '+' denotes LysoTracker co-localisation with Mtb, '−' denotes no co-localisation. Scale bar, 1 μm. **e** $P_{tcrXY}$ activity in Mtb H37Rv following the infection of resting or activated murine BMDMs for 24 h. Data is presented as a sum Mtb reporter signal (mCherry signal in GFP-positive bacteria) per voxel obtained from two biological replicates, filtered by LysoTracker co-localisation in Imaris (v9.6.1). $n = 127$ (red), $n = 172$ (blue), $n = 71$ (green), $n = 214$ (purple). Horizontal bars denote the median. Statistical analysis was performed using a one-way ANOVA with Turkey's multiple comparisons test. **f** Activity of indicated promoter-reporter constructs in Mtb H37Rv (WT) or a Δ*tcrXY* mutant after 24 h. Each bar represents the median mCherry fluorescence intensity from three replicates (open circle), representative of two biological replicates. Statistical analysis was performed using an unpaired two-tailed *t*-test. **g** Mutant alleles (*tcrX*$^{D54A}$*Y* or *tcrXY*$^{H256Q}$) or a wild-type allele (*tcrXY*) were expressed in a Δ*tcrXY* genetic background. Strains were cultured in pH 7 or pH 5.4 media for 24 h and *tcrX* mRNA levels were compared by qRT-PCR. H37Rv (WT) mRNA levels were used as a control. Each bar represents the mean fold-change from three biological replicates (open circle). Statistical analysis was performed using an unpaired two-tailed *t*-test. **h** Schematic of TcrXY signal transduction and autoregulation in Mtb at acidic pH. Image was created using BioRender.com. All error bars denote SD. Source data are provided as a Source Data file.

conserved hypotheticals, *Rv3706c* and *Rv3705A*, that are predicted to encode two small proline-rich proteins (SPRPs)[27]. These genes are conserved amongst the Mtb complex and notable non-tuberculous mycobacteria (Supplementary Fig. 6). We confirmed

pH- and TcrXY-dependent differential expression by qRT-PCR (Supplementary Fig. 7).

To assess whether the transcription of *Rv3706c* and *Rv3705A* is directly regulated by TcrX, we extracted ~300 bp sequences

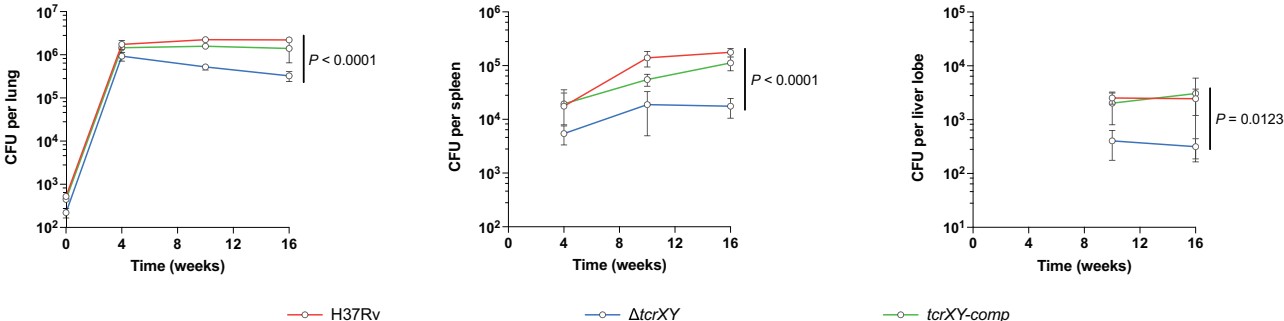

**Fig. 2 | The loss of TcrXY signalling attenuates Mtb survival during chronic infection.** Growth and survival of the Δ*tcrXY* mutant in the lungs, spleens, and livers of female C57BL/6 mice following low-dose aerosol infection. The infectious dose was confirmed from two mice from each group on day 1. Each time point represents the mean recovered CFU from each organ from five mice (except week 16 which represents four mice in the Δ*tcrXY* group), with error bars denoting SD. Statistical analysis was performed using an unpaired two-tailed *t*-test at week 16. Source data are provided as a Source Data file.

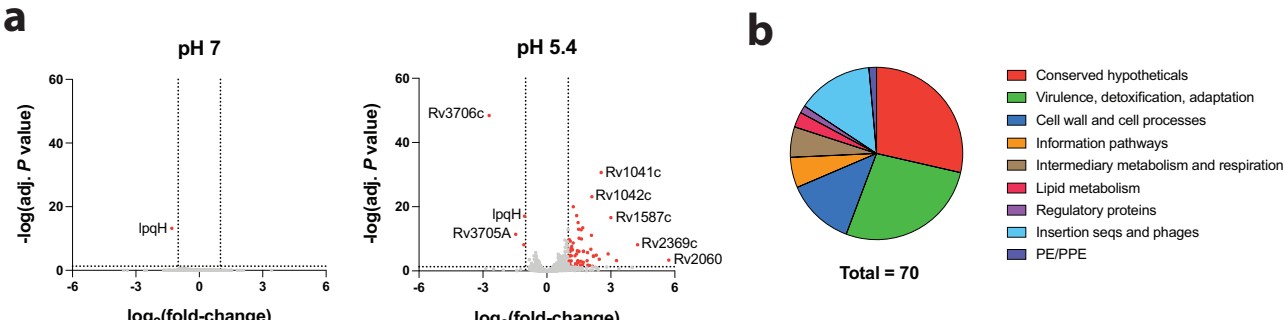

**Fig. 3 | Impact of TcrXY deletion on Mtb transcriptional adaptation to acidic pH. a** Volcano plot of all differentially regulated genes at pH 7 or pH 5.4 in a Δ*tcrXY* background relative to H37Rv. Each point represents the mean log$_2$(fold-change) and -log(adj. *P* value) across three biological replicates. Significantly differentially expressed genes are defined as fold-change >2 and adj. *P* < 0.05 (red). Genes not significantly differentially expressed are shown in grey. *P* values were adjusted for multiple testing using the Benjamini–Hochberg method[26]. Complete gene lists are supplied in Supplementary Data 1. **b** Functional categories of genes dysregulated in a Δ*tcrXY* background at acidic pH (*n* = 70). Gene functional categories were acquired using MycoBrowser[27]. Source data are provided as a Source Data file.

immediately upstream of their translational start sites and analysed their interactions using an EMSA (Supplementary Fig. 8). Purified TcrX demonstrated a direct interaction with P$_{Rv3706c}$ and P$_{Rv3705A}$, which was enhanced upon in vitro TcrX phosphorylation (Supplementary Fig. 8b, c). We examined the specificities of these interactions using unlabelled specific and non-specific competitor DNA (Supplementary Fig. 8d, e). These results indicate that *Rv3706c* and *Rv3705A* are bona fide members of the TcrXY regulon. Interestingly, the promoter regions of these genes do not contain the TcrX binding motif, as is present in the *tcrXY* promoter. Consequently, we devised and conducted a bespoke genome-wide in silico screen, without revealing any obvious binding motif consensus. While more work is required to elucidate promoter-response regulator interactions, it appears that the mechanism employed by TcrX may be atypical in relation to its downstream regulon genes.

To contextualise the activation of TcrXY signalling, we chose to further characterise the regulation of *Rv3706c*, since this gene was the most significantly differentially expressed in our analysis. We subsequently developed a Mtb dual transcriptional reporter to monitor *Rv3706c* transcription. This Mtb *Rv3706c* reporter displayed pH- and time-dependent activation in culture (~2-fold) which was completely dependent on the TcrXY TCS (Fig. 4a). Next, we assessed whether the pH-dependent activation of *Rv3706c* transcription in culture translated into similar responses during the infection of macrophages. Resting or activated murine BMDMs were infected with H37Rv (P$_{Rv3706c}$::*mCherry*; P$_{rpoB}$::*gfp*) or Δ*tcrXY* (P$_{Rv3706c}$::*mCherry*; P$_{rpoB}$::*gfp*) for 24 h and acidic compartments stained with LysoTracker (Fig. 4b). In H37Rv-infected

resting or activated BMDMs, a significant increase in *Rv3706c* reporter signal was observed in bacteria residing in acidified compartments (Fig. 4c). Induction of *Rv3706c* reporter signal in acidic compartments was completely abolished in a Δ*tcrXY* background. To establish the direct contribution of TcrXY signalling for *Rv3706c* induction, we utilised our TcrX/TcrY phosphorylation mutants. Induction of *Rv3706c* mRNA at acidic pH was completely abolished in these strains as compared to our phosphorylation-intact strains (~2-fold upregulation) (Fig. 4d). Together, these data support TcrXY as an acid-sensing TCS that directly activates the transcription of *Rv3706c* and *Rv3705A*.

### Rv3706c and Rv3705A mitigate redox stress at acidic pH

Inducible CRISPRi developed by Rock and colleagues[28] for use with Mtb is a powerful tool for investigating Mtb biology and pathogenesis, allowing for the rapid interrogation of newly identified genes of interest, without the time-consuming need for gene deleted and complemented strain generation. To assess the importance of *Rv3706c* and *Rv3705A* during Mtb infection, we generated two inducible CRISPRi silencing strains of H37Rv, designated variant by inducible CRISPRi expression (VICE)-*Rv3706c* and VICE-*Rv3705A*. Both VICE strains displayed sensitive and robust target gene knockdown by qRT-PCR (10-fold repression) (Supplementary Fig. 9). In activated BMDMs, induction of CRISPRi attenuated the intracellular survival of both Mtb strains by 60% after 72 h (Fig. 5a). Complementation with a CRISPRi-resistant allele of *Rv3706c* or *Rv3705A* reverted the survival defect. This indicates that these genes play an important, previously unknown, role in Mtb survival and persistence.

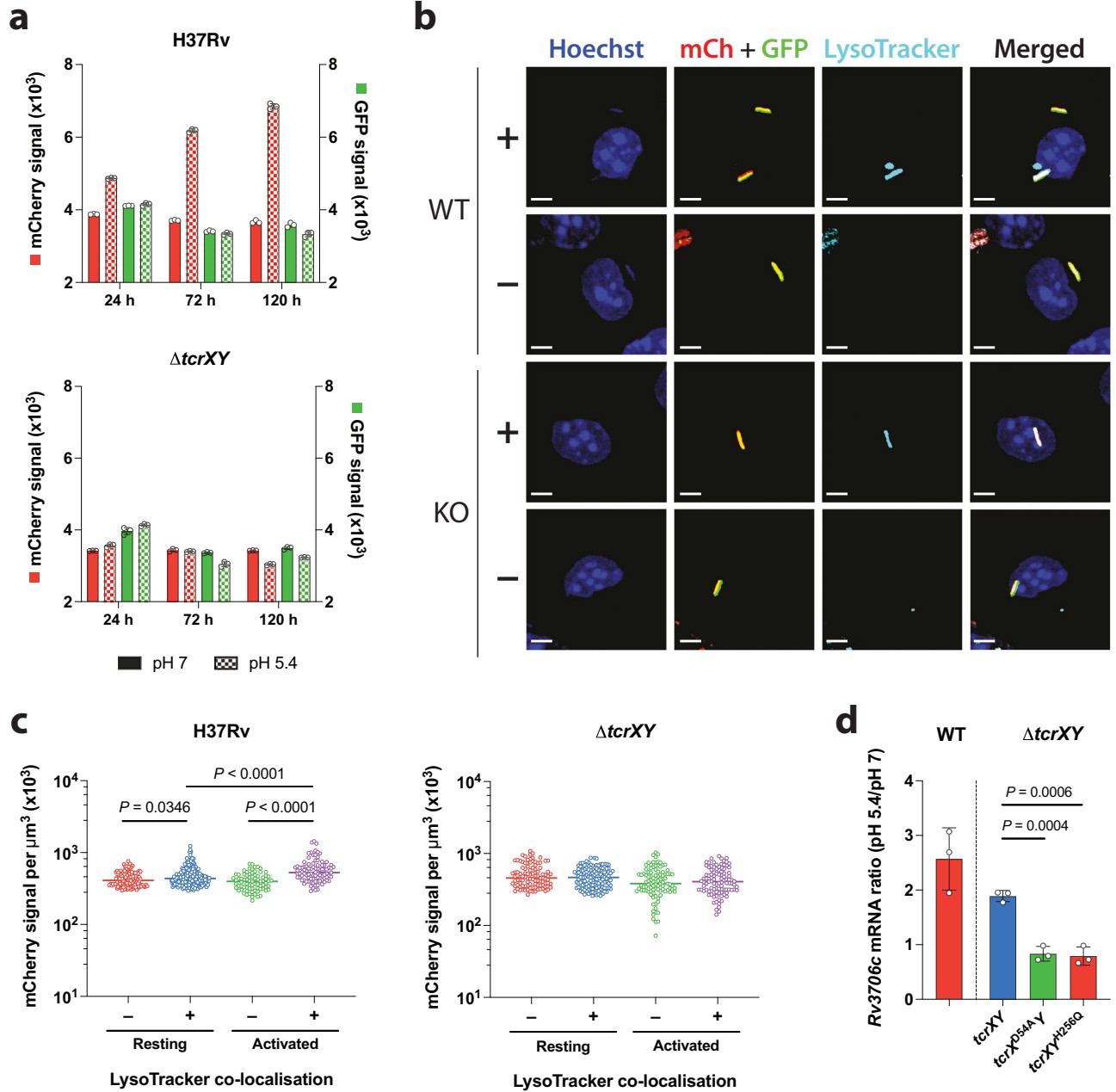

**Fig. 4 | TcrXY-dependent gene expression is activated at acidic pH. a** TcrXY is required for *Rv3706c* induction at acidic pH. H37Rv (P$_{Rv3706c}$::*mCherry*; P$_{rpoB}$::*gfp*) and Δ*tcrXY* (P$_{Rv3706c}$::*mCherry*; P$_{rpoB}$::*gfp*) were cultured in pH 7 or pH 5.4 minimal media, then mCherry and GFP fluorescence quantified by flow cytometry. Each bar represents the median fluorescence intensity from three replicates (open circle), representing two experiments. Error bars denote SD. **b** Exemplar confocal micrographs of resting murine BMDMs infected with H37Rv (P$_{Rv3706c}$::*mCherry*; P$_{rpoB}$::*gfp*) or Δ*tcrXY* (P$_{Rv3706c}$::*mCherry*; P$_{rpoB}$::*gfp*) for 24 h and stained with LysoTracker. Images depict cell nucleus (blue, Hoechst), H37Rv (WT) or Δ*tcrXY* (KO) reporter strain (red, mCherry (mCh) and green, GFP) and acidic compartments (cyan, LysoTracker). '+' denotes LysoTracker co-localisation with Mtb, '−' denotes no co-localisation. Scale bar, 2 μm. **c** *Rv3706c* transcription is activated in acidic intracellular compartments in a TcrXY-dependent manner. Resting or activated BMDMs were infected with H37Rv (P$_{Rv3706c}$::*mCherry*; P$_{rpoB}$::*gfp*) or Δ*tcrXY* (P$_{Rv3706c}$::*mCherry*; P$_{rpoB}$::*gfp*) for 24 h and fluorescence quantified by microscopy.

Data is presented as sum Mtb reporter signal (mCherry signal in GFP-positive bacteria) per voxel obtained from two biological replicates, filtered by LysoTracker co-localisation in Imaris (v9.6.1). For H37Rv-infected BMDMs, $n = 140$ (red), $n = 175$ (blue), $n = 78$ (green), $n = 93$ (purple). For Δ*tcrXY*-infected BMDMs, $n = 104$ (red), $n = 133$ (blue), $n = 101$ (green), $n = 101$ (purple). Horizontal bars denote the median. Statistical analysis was performed using a one-way ANOVA with Turkey's multiple comparisons test. **d** TcrXY signalling is required for induction of *Rv3706c* during acid stress. A wild-type *tcrXY* allele, phospho-mutant *tcrX*$^{D54AY}$ or *tcrXY*$^{H256Q}$ allele, were expressed in a Δ*tcrXY* genetic background. Strains were cultured at pH 7 or pH 5.4 for 24 h and *Rv3706c* mRNA levels were compared by qRT-PCR. H37Rv (WT) mRNA levels were used as a control. Each bar represents the mean fold-change from three biological replicates (open circle). Statistical analysis was performed using an unpaired two-tailed *t*-test. Error bars denote SD. Source data are provided as a Source Data file.

To gain insight into their biological function, we assessed the impact of *Rv3706c* and *Rv3705A* silencing on the Mtb proteome using unlabelled sequential window acquisition of all theoretical ions mass spectrometry (SWATH-MS). We cultured VICE-*Rv3706c* and VICE-*Rv3705A* at neutral or acidic pH in the presence or absence of ATc for 5 days and analysed whole cell extracts. Over 3000 proteins were identified and quantified across all samples, representing approximately 75% coverage of the H37Rv proteome. Silencing of

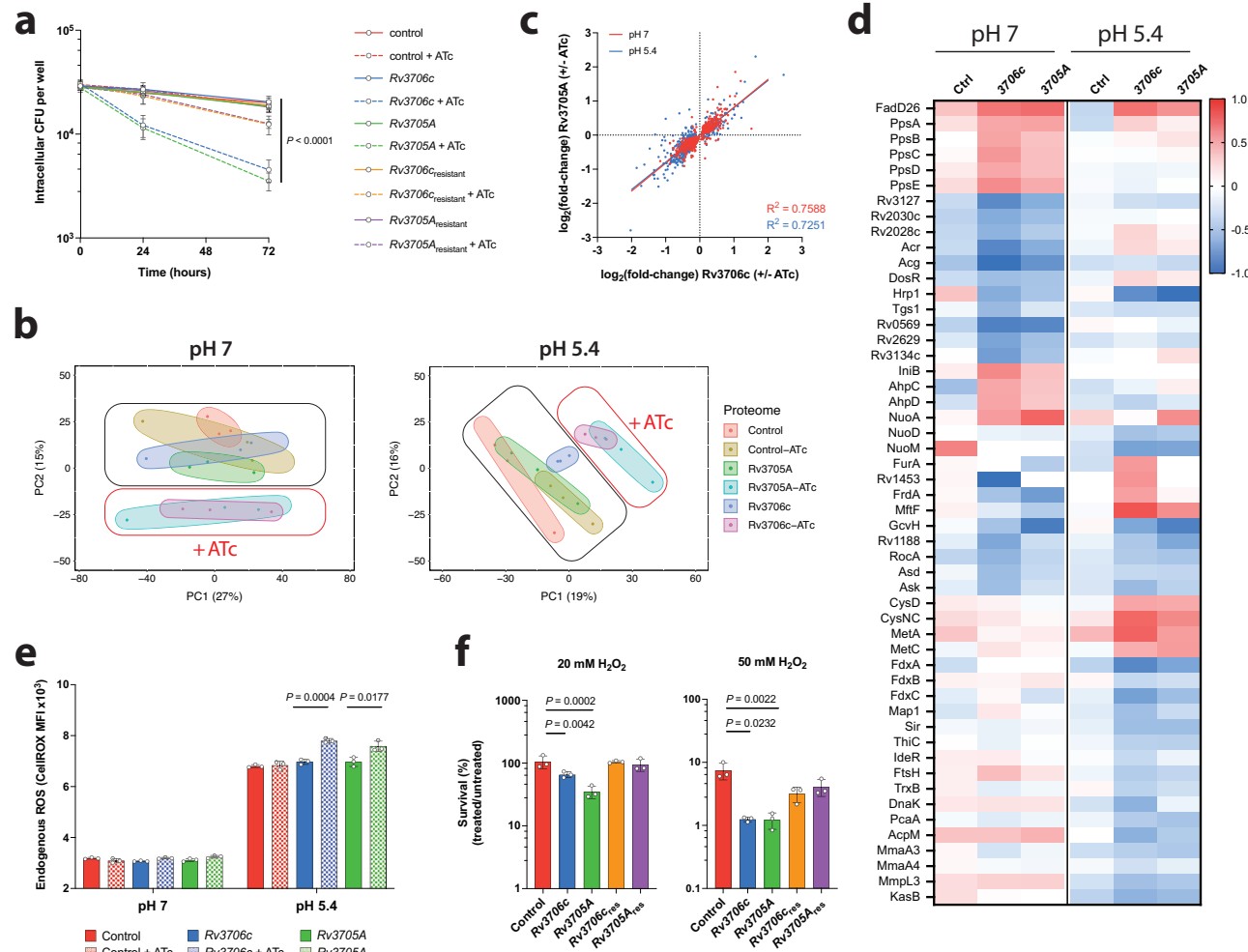

**Fig. 5 | Rv3706c and Rv3705A mitigate redox stress to permit Mtb persistence.**
**a** Mtb intracellular survival in activated murine BMDMs is attenuated upon CRISPRi-mediated silencing of *Rv3706c* or *Rv3705A*. VICE strains were pre-induced with 200 ng/mL ATc for 72 h prior to infection, and ATc was maintained in the culture medium for the duration of the infection. Each time point represents the mean recovered CFU of two biological replicates each performed in triplicate, with error bars denoting SD. Statistical analysis of VICE strains with or without ATc was performed using an unpaired two-tailed *t*-test at 72 h. **b** Principal component analysis (PCA) of the Mtb proteome upon CRISPRi-mediated gene silencing of *Rv3706c* and *Rv3705A* reveals significant remodelling of physiology. **c** Linear regression analysis of the dysregulated Mtb proteome reveals significant overlap upon *Rv3706c* and *Rv3705A* gene silencing. Each point represents a significantly differentially abundant protein (adj. $P < 10^{-5}$) in each respective comparison (genotype ± ATc). Line of best fit is shown in each comparison (pH 7 and pH 5.4). In each comparison, $P < 0.001$. **d** Genetic silencing of *Rv3706c* and *Rv3705A* alters several biological pathways including redox homoeostasis in Mtb. Heatmap comparing differential

protein abundance as $\log_2$(fold-change) of each VICE strain, and a control VICE strain (Ctrl), calculated from three biological replicates. All proteins are below the statistical significance cut-off (adj. $P < 10^{-5}$). Complete protein lists are supplied in Supplementary Data 2 and 3. **e** Depletion of Rv3706c or Rv3705A in Mtb promotes the accumulation of endogenous reactive oxygen species (ROS). VICE strains were grown in 7H9 broth at the indicated pH in the presence or absence of ATc and then stained with the fluorescent dye CellROX Red for 2 h. Each bar represents the median fluorescence intensity (MFI) from three independent cultures (open circles). Statistical analysis was performed using an unpaired two-tailed *t*-test. Error bars denote SD. **f** CRISPRi-mediated silencing of *Rv3706c* or *Rv3705A* enhances Mtb sensitivity to oxidative stress at acidic pH. Each bar represents mean survival of each VICE strain from three independent cultures (open circle), relative to an unchallenged control assayed in parallel. Error bars denote SD. Statistical analysis was performed using an unpaired two-tailed *t*-test. Source data are provided as a Source Data file.

Mtb *Rv3706c* and *Rv3705A* had significant and overlapping impacts on the proteome under both conditions (Fig. 5b). Importantly, a CRISPRi control strain (empty vector) displayed little impact on the proteome (Supplementary Data 2a, b and 3a, b). Given the stark similarity in proteomic profiles, we directly compared all significantly differentially abundant proteins in both genotypes by their magnitude difference and observed a highly significant correlation (Fig. 5c, Supplementary Data 2c–f and 3c–f). Functional pathway enrichment revealed significant remodelling of several metabolic and redox pathways (Supplementary Data 4 and 5). Under neutral conditions (pH 7), we observed global changes consistent with remodelling of the respiratory chain (NADH dehydrogenases and

DosR-regulated proteins) (Gene Ontology (GO): 0030964 and GO: 0050896), repression of amino acid catabolism (Rv1188, RocA, Asd, Ask) (GO: 0006520), enhanced lipid anabolism (PpsA-E) (GO: 0034081) and dysregulated redox metabolism (GO: 0016491 and GO: 1990204) in both VICE strains (Fig. 5d). At acidic pH, we observed changes consistent with counteracting oxidative stress (GO:0071451, GO:1901701 and GO:0006979), repression of amino acid metabolism (GO: 0006520), enhanced sulfur metabolism (CysD, CysNC, MetA, MetC) (GO: GO:0006790), and mycolic acid biosynthesis (GO: 0071767) (Fig. 5d). Together, these findings indicate that *Rv3706c* and *Rv3705A* have significant impacts on Mtb physiology.

Acidic pH alone in culture is sufficient to induce extensive redox stress and results in the accumulation of endogenous reactive oxygen species (ROS)[17,29,30]. We wondered whether Rv3706c and Rv3705A had an impact on endogenous ROS at acidic pH. Consistent with prior reports, acidic pH significantly increased endogenous ROS levels in Mtb (Fig. 5e). A further increase in endogenous ROS was observed upon CRISPRi-silencing of *Rv3706c* or *Rv3705A* at acidic pH. Further, we hypothesised that a shift in redox balance, as a result of Rv3706c or Rv3705A depletion, may sensitise Mtb to host-derived oxidative stress. We challenged acid-adapted Mtb with 20 or 50 mM hydrogen peroxide for 2 h and monitored the impact of genetic silencing. CRISPRi silencing of *Rv3706c* and *Rv3705A* reduced Mtb tolerance to 20 mM $H_2O_2$ challenge by 38% and 67%, respectively (Fig. 5f). Tolerance to 50 mM $H_2O_2$ challenge was reduced by 83% and 84%, respectively. Overall, these results show that Rv3706c and Rv3705A contribute to redox homoeostasis in Mtb and play a role in mitigating oxidative stress during infection to permit intracellular survival.

## Inhibition of TcrXY signalling attenuates persistence and accelerates bacterial clearance with front-line anti-TB drugs

We sought to assess whether TcrXY-dependent acid sensing and response may represent a target vulnerable to in vivo inhibition. To specifically deplete TcrX during chronic infection, we constructed an inducible CRISPRi-mediated *tcrX* knockdown mutant, VICE-*tcrX*, capable of sensitive and robust *tcrX* silencing in Mtb (Supplementary Fig. 10a, b). In mice, silencing of *tcrX* four weeks post-infection, i.e., at the onset of chronic infection, significantly attenuated Mtb persistence by more than 10-fold in the lungs and spleen (competitive index (CI) of 0.086 and 0.064, respectively) over a period of 10 weeks (Fig. 6a,b). Analogous to the Δ*tcrXY* knock-out mutant, silencing of *tcrX* using CRISPRi had no impact on Mtb growth in the lungs during acute infection (Supplementary Fig. 10c). These data indicate that inducible depletion of TcrX following the establishment of a chronic infection attenuates Mtb persistence.

Next, we asked if TcrXY inhibition could be leveraged to improve Mtb clearance in front-line treatment regimens. In chronically-infected

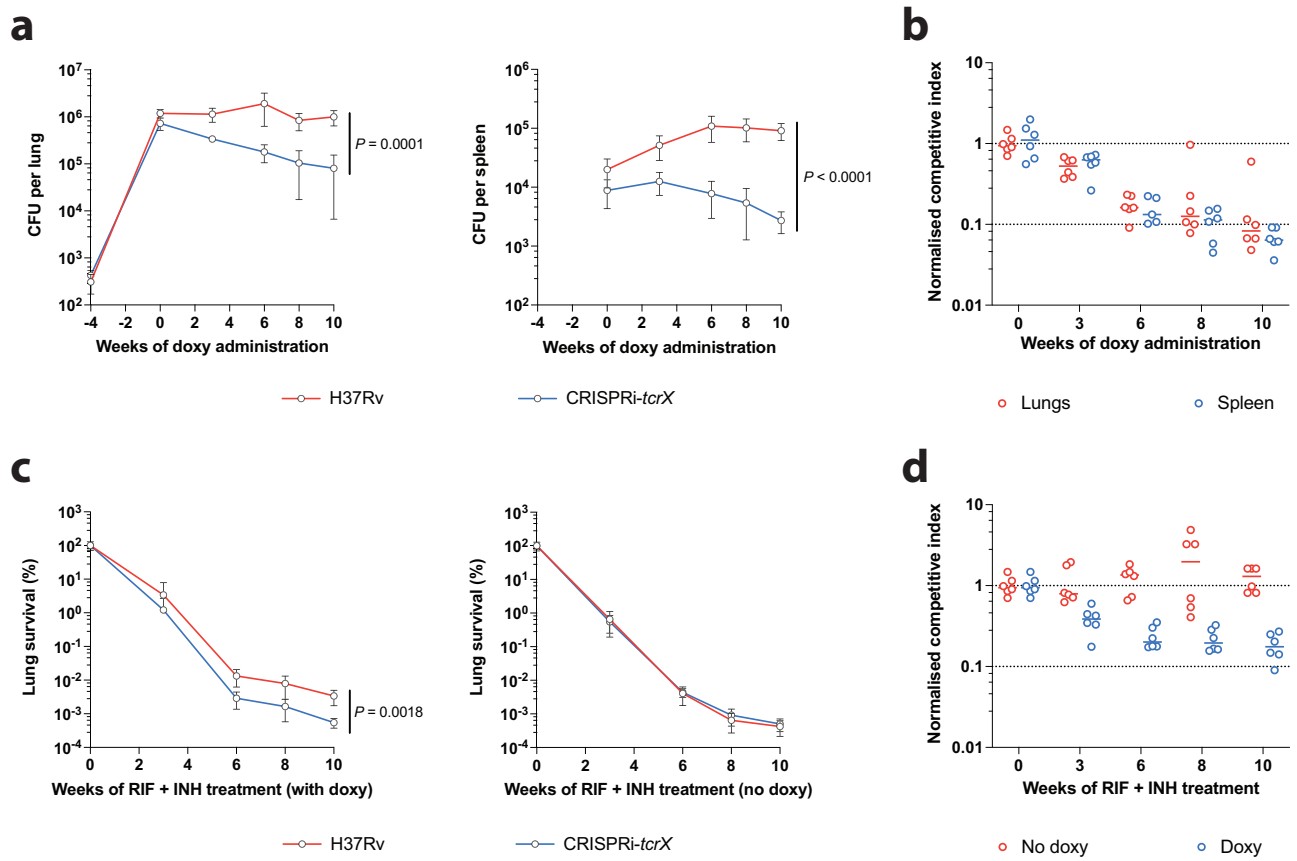

**Fig. 6 | Inhibition of the TcrXY TCS attenuates Mtb survival during chronic infection and accelerates bacterial clearance with front-line anti-TB agents.** **a** Mtb survival in the lungs and spleens of chronically-infected mice upon CRISPRi-mediated *tcrX* silencing. Female C57BL/6 mice were co-infected with <500 CFU of a 1:1 mix of Mtb H37Rv and CRISPRi-*tcrX*. Mice were rested for 4 weeks to establish a wild-type like chronic infection, before CRISPRi-mediated TcrX depletion by doxycycline (doxy) administration in mouse chow (2000 ppm). Each time point represents the mean recovered CFU from each organ from six mice, with error bars denoting SD. Statistical analysis was performed using an unpaired two-tailed *t*-test at week 10. **b** Mtb fitness in the lungs and spleens of chronically-infected mice upon CRISPRi-mediated *tcrX* silencing. Competitive index (CI) from (**a**) was calculated from recovered CFU counts for each strain, and normalised to the onset of doxy administration. Each point is the CI from each mouse organ, with the median of each group denoted by a horizontal line. **c**, Mtb clearance from the lungs of chronically-infected mice treated with rifampicin (RIF) and isoniazid (INH). Female

C57BL/6 mice were co-infected with <500 CFU of a 1:1 mix of Mtb H37Rv and CRISPRi-*tcrX*. Mice were rested for 4 weeks to establish a wild-type like chronic infection, before treatment with RIF and INH (0.1 mg/mL in drinking water). CRISPRi-mediated TcrX depletion was induced by doxy administration in mouse chow (2000 ppm) (with doxy). In parallel, a control group received the same base mouse feed composition without doxy (no doxy). Each time point represents the mean recovered CFU from each organ from six mice, with error bars denoting SD. Statistical analysis was performed using an unpaired two-tailed *t*-test at 10 weeks of treatment. **d** Mtb fitness in the lungs of chronically-infected mice treated with RIF and INH upon CRISPRi-mediated *tcrX* silencing. Competitive index from (**c**) was calculated from recovered CFU counts for each strain, and normalised to the onset of doxy administration. Each point is the CI from each mouse lung, with the median of each group denoted by a horizontal line. Source data are provided as a Source Data file.

mice treated with rifampicin and isoniazid for 10 weeks, Mtb survival in the lungs was further arrested upon silencing of *tcrX* (CI = 0.175) (Fig. 6c, d). Genetic silencing of *tcrX* reduced bacterial numbers approximately 10-fold compared to drug treatment groups alone. Importantly, in the absence of CRISPRi-mediated silencing, Mtb fitness and survival in vivo were unaltered (CI = -1) (Fig. 6c, d). This is significant because bacterial clearance with front-line agents is drastically reduced in chronically-infected mice[31,32]. Mtb sensitivity to rifampicin and isoniazid in vitro was unaffected by *tcrX* silencing (Supplementary Fig. 11). These data suggest that the loss of acid-sensing/response in Mtb via disruption of TcrXY signalling may be used to improve bacterial clearance in front-line regimens.

## Discussion

A growing body of research supports acidic pH as an important host-derived pressure in mycobacterial infection biology. In its lifecycle, Mtb encounters environments of varying acidity, including the host cell cytoplasm (-pH 7)[33], phagosome (-pH 6)[22], phagolysosome (-5.5–5)[22,34] and granuloma (median pH 5.5, range pH 7.2–5)[35]. Evidence that Mtb perceives acidity during infection and coordinates an adaptive response is apparent in its expansive transcriptional response to the stimulus[11,20]. Since Mtb maintains a near-neutral cytoplasm at acidic pH[16], the pathogen clearly requires mechanisms to sense this extra-cytoplasmic environmental stimulus to achieve essential adaptive responses. The TcrXY TCS affords such a mechanism. Our findings support TcrXY as a pH sensor in Mtb, translating acidic pH to an internal response via phosphorylation signalling and genetic regulation. Intramacrophage pH triggers TcrY-TcrX signal transduction to co-ordinate a 70-gene regulon required for effective persistence during the chronic infection of mice.

Interestingly, TcrXY is not the only Mtb TCS with documented pH sensing capacity. The PhoPR regulon is similarly induced in response to intramacrophage pH stress[11,30,36], along with other stresses[37,38], to control a 114-gene regulon[39]. We observed little overlap between the two regulons (-2% of genes[38,39]), indicating that their regulatory outputs are largely distinct. With this in mind, it is intriguing to speculate how Mtb co-ordinates multiple pH-sensitive signalling pathways during infection. One view is that the TCSs may respond optimally to different ranges of pH. The PhoPR regulon is induced rapidly following infection of naïve macrophages[20], and is unaffected by further acidification of the phagosome[36], indicating that it is sensitive to subtle changes in pH that occur early following colonisation[36,37]. In contrast, the TcrXY system appears to be more important for responding to more severe pH stress that occurs later in infection. This is reflected in the two regulons, with PhoPR regulating genes associated with early phagosomal damage (e.g., ESX-1 secretion[40,41]) and TcrXY regulating genes associated with long-term persistence (e.g., toxin-antitoxin systems). Overall, our study and others support the existence of multiple pH-sensitive signalling pathways in Mtb, highlighting both the complexity and nuance of the host-pathogen interaction and the need for the organism to be able to respond to a gradient of pH stress in order to establish and maintain infection.

We provide here a description of two TcrXY regulon members, *Rv3706c* and *Rv3705A*, both annotated as putative proline-rich proteins[27]. In host cells, Mtb experiences extensive redox pressure[42,43] (e.g., via inhibited respiration, acidic pH, host-derived oxidants, and fatty acid metabolism). Acidic pH alone in culture is sufficient to induce redox imbalances and upregulate redox-sensitive pathways[11,29], and this is likely compounded by other host-derived pressures. We show here that during the infection of macrophages, *Rv3706c* and *Rv3705A* are induced via TcrXY signalling to mitigate redox stress and promote Mtb survival. We therefore propose a renaming of *Rv3706c* and *Rv3705A* to better represent their biological function; <u>T</u>crX and <u>a</u>cid-regulated <u>r</u>edox homoeostasis genes A and B (*tarA* and *tarB*, respectively). While the mechanism

responsible for TarA/TarB-mediated protection against oxidative stress remains unknown, it is alluring to draw parallels with an emerging field of study – the small proline-rich protein (SPRP) family in higher eukaryotes[44]. SPRPs are prevalent in diverse tissues[45], and are induced in response to inflammation and ROS-mediated damage[46,47]. The coding sequences of SPRPs in higher eukaryotes are enriched in free radical quenching cysteine, proline and histidine[48,49]. In polypeptides, exposure of prolyl residues to hydroxyl free radicals results in hydroxylation of the pyrrolidine ring to generate 3- and 4-hydroxyproline[50–53] (i.e., a non-enzymatic scavenging mechanism). In contrast to animals, plants, and fungi, the potential of free and polypeptide-bound proline to act as a ROS-detoxification mechanism in bacterial pathogens is poorly studied. We speculate that TarA and TarB may sequester endogenous ROS associated with acidic pH or other host-derived stress in Mtb via the formation of proline-hydroxyl adducts. The greater protection afforded by TarB to $H_2O_2$ challenge may reflect the greater number of proline residues in the nascent polypeptide (30 in TarB versus 18 in TarA, representing 23% and 17% of their ORF respectively).

In this study, we put forward the hypothesis that TcrXY signalling could represent a vulnerability during chronic infection. We reasoned that targeting this pathway could potentially render persistent populations of Mtb susceptible to immune clearance and/or antibiotic-mediated killing, ultimately leading to improved treatment outcomes and shorter therapy durations. To address this hypothesis, we leveraged the newly developed CRISPRi platform[28] in Mtb to specifically deplete TcrX in chronically infected mice, thereby mimicking the effects of a specific inhibitor. We found CRISPRi to be highly suitable for this purpose, providing the flexibility to induce target depletion following the onset of chronic infection and the specificity to phenocopy our knockout mutant in mice. The inhibition of TcrXY signalling attenuated the fitness of Mtb in the lungs of chronically-infected mice, and accelerated clearance of Mtb in mice treated with front-line anti-TB drugs. Our work thus endorses future efforts to develop inhibitors of this signalling pathway and supports the use of CRISPRi to evaluate the viability of targets of interest in the drug discovery field. A slight reduction in clearance rates of Mtb was observed in RIF/INH treated mice when fed doxycycline-containing chow (Fig. 6c). It is possible that doxycycline reduces the palatability of the chow, as it does when added to drinking water, which requires the addition of sucrose to encourage consumption. A reduction in food intake may lead to a reduced intake of drinking water and hence antibiotic dosage. While this effect was only mild and does not impact the interpretation of outcomes reported, it should be considered when conducting doxycycline-induced CRISPRi gene silencing in vivo.

In recent years, TCS inhibitors have gained significant attention as an approach to disarm in vivo virulence/persistence attributes in a variety of bacterial pathogens including Mtb[38,54–57]. The architecture of TCS signalling in prokaryotes presents several vulnerabilities, which include signal sensing and autophosphorylation of the sensor kinase, phosphorelay and activation of the response regulator, and the DNA-binding activities of the regulator. In this sense, the likelihood of identifying inhibitors of TCS signalling is potentially far greater than conventional enzymatic targets given the capacity of a small molecule to elicit the desired phenotype through several modes of action along the pathway. Further, since TCSs are not found in mammalian cells[58], they represent attractive drug targets as antagonists of the signalling pathway and would have limited potential off-target effects. The conservation of TCS architecture also posits the possibility to target several TCSs simultaneously in the pathogen[58–60], thus providing global disruption of stimulus sensing and response. Overall, our study supports targeting the TcrXY TCS as an adjunctive approach toward this goal, to attenuate Mtb persistence and potentially shorten front-line therapy duration.

## Methods

### Ethics statement
All experiments involving the production, and use, of genetically modified bacterial strains were approved by The University of Queensland Institutional Biosafety Committee (IBC) in accordance with legislation and guidelines as per the Office of the Gene Technology Regulator, Australia. All experiments requiring *Mycobacterium tuberculosis* were performed in an IBC-approved physical containment level 3 (PC3) facility at The University of Queensland, providing a 12-h light/dark cycle, ambient temperature range of 22–24 °C, at 40–60% humidity. The Animal Ethics Committee of The University of Queensland approved all experiments involving mice under application SCMB/128/19.

### Bacterial strains and growth conditions
*Mycobacterium tuberculosis* H37Rv was used as the parental strain throughout this study. Bacteria were routinely cultured using Middlebrook 7H9 liquid medium supplemented with 0.2% glycerol, 0.02% tyloxapol, and 10% oleic acid-albumin-dextrose-catalase (OADC) or Middlebrook 7H10 solid agar supplemented with 0.2% glycerol and 10% OADC. Antibiotics for the selection of genetically modified strains were added as required: kanamycin (25 µg/mL), hygromycin (50 µg/mL), and/or nourseothricin (25 µg/mL). CRISPRi was induced using 200 ng/mL of anhydrotetracycline (ATc). Mtb cultures were grown statically at 37 °C. All strains were stored in 15% glycerol at −80 °C.

All acid stress assays were performed in a defined minimal medium (1 g/L $KH_2PO_4$, 2.5 g/L $Na_2PO_4$, 0.5 g/L $(NH_4)_2SO_4$, 0.15 g/L asparagine, 10 mg/L $MgSO_4$, 50 mg/L ferric ammonium citrate, 0.1 mg/L $ZnSO_4$, 0.5 mg/L $CaCl_2$, 0.02% tyloxapol and 0.2% glycerol) as described previously[61]. Media was buffered using 100 mM MOPS (pH 7), 100 mM MES (pH 6.5–5.4), or 100 mM phosphate-citrate (pH 5–4). Following 4 weeks of Mtb culture, no significant changes in the pH of the medium were observed (<0.1 units), indicating effective buffering of the growth medium.

### Mice
Female C57BL/6 mice were sourced from the Animal Resources Centre (Perth, Australia). All mice were housed in a Techniplast IVC system with unrestricted access to food and water. Mice were used at 6-8 weeks of age.

### DNA cloning and analysis
Cloning of DNA constructs was performed using Homologous Alignment Cloning[62]. A 20 µL reaction was prepared containing 100 ng of linearised vector DNA, insert DNA (3:1 insert:vector molar ratio), and 1 U of T4 DNA polymerase (Thermo Fisher) in 1X T4 reaction buffer (Thermo Fisher). The reaction was incubated at 37 °C for 1 min and stopped by the addition of 1 µL of 0.5 M EDTA (pH 5). The mixture was then incubated at 60 °C for 1 min and allowed to cool slowly to room temp, allowing for efficient annealing of complementary vector-insert overhangs[62]. The resulting recombinant molecules were transformed into chemically competent *E. coli* DH5α prepared in-house using the TSS method[63]. A full list of plasmids used in this study is available in Supplementary Table 1. A full list of oligonucleotides used in this study is available in Supplementary Data 6. All constructs were confirmed by DNA sequencing.

### Mutant generation and complementation
The Δ*tcrXY* mutant was constructed by replacing the entire *tcrX* (*Rv3765c*) and *tcrY* (*Rv3764c*) open reading frame with a hygromycin-resistance cassette in Mtb using recombineering[64]. To achieve this, H37Rv transformed with pJV53 was grown to mid-log phase and treated with 200 mM glycine. After 16 h, mycobacterial recombinases were induced with 0.2% acetamide for a further 24 h prior to making cells electrocompetent. A DNA fragment containing the hygromycin-resistance cassette flanked by 1-kb of the upstream sequence from *tcrX* and 1-kb of the downstream sequence from *tcrY* was transformed by electroporation. Knockout candidates were selected on hygromycin-containing agar and screened for homologous recombination by PCR. A complemented strain, *tcrXY*-comp, was obtained by introducing a chromosomal copy of wild-type *tcrXY* (under the control of the putative *tcrXY* promoter ($P_{tcrXY}$)) into the Δ*tcrXY* genome. Complementation strains expressing *tcrXY* harbouring phosphorylation-ablative point mutations (*tcrX*$^{D54A}$; *tcrY*$^{H256Q}$) were generated by site-directed mutagenesis and introduced into the Δ*tcrXY* genome as above.

### Cloning of VICE (Variant by Inducible CRISPRi Expression) strains and complement generation
CRISPRi strains (VICE) were constructed essentially as described[65]. Briefly, the CRISPRi plasmid backbone (PLJR965; Addgene plasmid #115163) was digested with BsmBI-v2 (New England Biolabs) and gel purified. For each individual sgRNA, two complementary oligonucleotides were annealed and cloned into the digested plasmid backbone. When indicated, complementation of CRISPRi-mediated gene knockdown was achieved by introducing synonymous mutations into the native allele and the protospacer adjacent motif (PAM) to inhibit sgRNA targeting. CRISPRi-resistant alleles were synthesised as a custom gene block (Integrated DNA Technologies) and expressed from their endogenous promoter in a Giles integrating plasmid, transformed into the respective VICE strain.

### Reporter generation and in vitro assays
To generate H37Rv ($P_{tcrXY}$::*mCherry*; $P_{rpoB}$::*gfp*) and Δ*tcrXY* ($P_{tcrXY}$::*mCherry*; $P_{rpoB}$::*gfp*), a 569-bp region immediately upstream of the *tcrX* start codon was transcriptionally coupled to *mCherry* in the integrative plasmid pDual306 or extrachromosomal plasmid pDual206 (see Supplementary Methods), and transformed into the respective genetic background by electroporation. To generate sequence-scrambled $P_{tcrXY}$ variants, a 7-bp mirror repeat in the $P_{tcrXY}$ DNA fragment was scrambled and cloned into pDual306 as above. To generate H37Rv ($P_{Rv3706c}$::*mCherry*; $P_{rpoB}$::*gfp*) and Δ*tcrXY* ($P_{Rv3706c}$::*mCherry*; $P_{rpoB}$::*gfp*), a 459-bp region immediately upstream of the *Rv3706c* start codon was transcriptionally coupled to *mCherry* in plasmid pDual306 or pDual206, and transformed into the respective genetic background by electroporation.

For broth assays, reporter strains were grown to mid-log phase in 7H9 complete, harvested by centrifugation, and then resuspended in buffered minimal media at a final $OD_{600}$ of 0.1. At desired time-points, bacteria were harvested by centrifugation, resuspended in 4% paraformaldehyde (PFA) in PBS and fixed at 4 °C overnight. Fixed cells were pelleted by centrifugation and resuspended in 1 mL of PBS-0.05% tyloxapol. Samples were immediately analysed using a CytoFLEX flow cytometer (Beckman Coulter).

### Macrophage infections
Murine bone marrow-derived macrophages (BMDM) were derived from bone marrow progenitors isolated from the femurs and tibias of C57BL/6 mice. Bones were flushed with sterile PBS, and bone marrow cells differentiated in RPMI-1640 medium (Thermo Fisher) supplemented with 10% foetal bovine serum (Bovogen), 0.015% sodium pyruvate (Thermo Fisher), 2 mM GlutaMAX (Thermo Fisher), 100 U/mL penicillin–streptomycin (Thermo Fisher) and 20 ng/mL recombinant macrophage colony-stimulating factor (m-CSF) (Peprotech) for 6 days at 37 °C, 5% $CO_2$. Recombinant m-CSF was replenished in the culture medium every 3 days. Differentiated BMDMs were activated with 200 units/mL recombinant IFN-γ (Peprotech) and 50 ng/mL lipopolysaccharide (LPS) (Thermo Fisher) for 16 h.

For infection assays, Mtb cultures were synchronised to the mid-log phase in 7H9 complete media. Mtb VICE strains were pre-induced with ATc 3 days prior to infection. Bacteria were resuspended in cell culture media, and declumped by vortexing with 1 mm glass beads (Sigma Aldrich) for 90 s and passing bacteria through a 5 μm PVDF filter (Merck). This produced a highly homogenous single-cell inoculum, as confirmed by microscopy. Cell culture medium was removed from the macrophages and replaced with Mtb-containing medium at a multiplicity of infection of 1. After 4 h of infection, the medium was removed, and cells were washed thrice with PBS and replenished with fresh cell culture media containing 10 ng/mL m-CSF. For CRISPRi infections, ATc was added to the media at 200 ng/mL to maintain target gene knockdown. To quantify intracellular bacterial load, the infection medium was removed at indicated time-points and cells lysed in 0.2 mL of 0.1% Triton X-100 in PBS for 10 min. Serially diluted lysates were cultured on 7H10 agar at 37 °C until colonies were visible (3 weeks).

## Confocal microscopy and image analysis

Murine BMDMs were infected with Mtb reporter strains cultured on sterile 12 mm round glass coverslips. After 24 h, the infection medium was supplemented with 100 nM LysoTracker Deep Red (Thermo Fisher) for 1 h. Cell culture medium was removed and cells were fixed in 4% PFA for 1 h at room temperature protected from light. The fixed coverslips were washed thrice with PBS to remove residual PFA and then stained with Wheat Germ Agglutinin Alexa Fluor 647 (Thermo Fisher) and NucBlue Live ReadyProbes Reagent (Hoechst 33342) (Thermo Fisher) as per manufacturer's instructions. Coverslips were again washed thrice with PBS and then mounted onto slides containing Dako fluorescence mounting medium (Sigma Aldrich).

Slides were imaged with a Zeiss LSM 900 confocal microscope on a ×40 oil objective and collected as 0.5 μm z-stacks for analysis. Image analysis was performed in Imaris (v9.6.1) by reconstructing z-stacks into three-dimensional surface models. Co-localisation with Lyso-Tracker was performed using the surface–surface co-localisation function (0 μm or less). For quantification of reporter signals, the fluorescence voxel volume of each bacterium was measured via the GFP channel (i.e., constitutive control signal), and the corresponding sum of the mCherry signal as a measure of promoter activity. Settings for the mCherry and GFP channels were maintained during the imaging of slides between experimental sets to allow signal comparisons. Bacterial fluorescence was quantified in each experimental condition over two independent biological replicates.

## RNA extraction and qRT-PCR

Mtb (~$10^9$ CFU) was harvested by centrifugation and RNA transcripts stabilised using RNAprotect Bacteria Reagent (Qiagen) as per manufacturer's instructions. Protected bacterial pellets were resuspended in Buffer RLT (Qiagen), disrupted by bead-beating using 0.1 mm Zirconia beads in a BeadBug Homogeniser, and lysates mixed with chloroform (20% final concentration). The aqueous phase was isolated following centrifugation and total RNA was extracted using the RNeasy mini kit (Qiagen) as per the manufacturer's instructions. Residual contaminating gDNA was digested with DNase I (Ambion) and RNA was re-purified by ethanol precipitation. Total RNA purity and integrity were routinely assessed spectrophotometrically and by gel electrophoresis.

Complementary DNA (cDNA) was prepared from 2 μg of DNA-free total RNA using the Tetro cDNA synthesis kit (Meridian Bioscience) with random hexamers. cDNA levels were then measured by quantitative reverse-transcription PCR (qRT-PCR) on a ViiA7 light cycler (Applied Biosystems) using QuantiFast SYBR Green PCR mix (Qiagen). All qPCR primers were verified to be >80% efficient for their target and cDNA levels were experimentally validated to be within the working linear range of each assay. Signals were normalised to the housekeeping *rpoB* transcript. Data analysis was performed using the delta-delta Ct method[66]. All qPCR primers used in this study are available in Supplementary Data 6.

## RNA sequencing and analysis

H37Rv and the Δ*tcrXY* mutant were grown to mid-log phase in 7H9 media, pelleted by centrifugation, and then resuspended in the minimal medium at pH 7 or pH 5.4 at a final $OD_{600}$ of 0.4. Cultures were maintained at 37 °C for 24 h, and total RNA extracted as above. RNA samples were quantified using the Qubit RNA Broad Range Assay kit (Invitrogen) and quality control was performed using RNA tapes on a TapeStation 4200 (Agilent). Total RNA (50 ng) was used to prepare RNAseq libraries using Illumina Stranded Total RNA Prep with Ribo-Zero Plus (Illumina) as per the manufacturer's instructions. Fifteen PCR cycles were performed to amplify and add indexes and primer sequences for sequencing. On completion of library preparation, each library was quantified, and quality control was performed using the Quant-iT dsDNA HS Assay kit (Invitrogen) and D1000 HS tapes on a TapeStation 4200 (Agilent). The library was prepared for sequencing on the NovaSeq6000 (Illumina) using NovaSeq6000 SP kit v1.5 (2 × 150-bp paired-end chemistry), in the Australian Centre for Ecogenomics at The University of Queensland.

Analysis of RNAseq data was performed using Galaxy Australia[67]. All parameters were set as default values unless otherwise stated. Sequence read quality was evaluated using FastQC (v0.73) (https://www.bioinformatics.babraham.ac.uk/projects/fastqc/), and reads trimmed using Trimmomatic (v0.36)[68] (LEADING: 20, TRAILING: 20). Paired-end reads were mapped against the reference genome H37Rv (Refseq: NC_000962.3) with Bowtie2 (v2.4.5)[69], using the 'very sensitive local' preset. The number of fragments mapped to each gene was counted using FeatureCounts (v2.0.1)[70], with the following parameters: stranded (reverse), GFF feature type filter and gene identifier were set as coding sequence (CDS) and locus_tag, respectively, and summarisation was performed at the feature level. Only fragments with both reads aligned were counted, and only fragments with at least 25 bases of overlap with a CDS were counted. Fragments that mapped to more than one chromosomal location were discarded. Differential gene expression analysis was performed using DESeq2 (v1.40.2)[71]. Significantly differentially expressed genes were considered with an adjusted $P < 0.05$. Mapped fragments (raw BAM file) were visualised using Artemis[72]. The RNAseq analysis pipeline used in this study is publicly available (https://usegalaxy.org.au/u/mstup1/w/rnaseq-analysis).

## Mass spectrometry and analysis

Mtb VICE strains were grown to mid-log phase in 7H9 media, and then seeded into fresh 7H9 media buffered to pH 7 or pH 5.4 at a final $OD_{600}$ of 0.2 in the presence or absence of ATc for 5 days. Bacteria were harvested by centrifugation, and whole cell extracts were prepared by resuspending the cell pellet in lysis buffer (50 mM Tris–HCl pH 7.4, 8 M Urea, 5 mM DTT, Roche cOmplete protease inhibitors) and bead-beating in a BeadBug Homogeniser using 0.1 mm Zirconia beads. The lysate was clarified by centrifugation (20,000×*g*, 15 min) and filtered through a 0.22 μm PVDF filter (Merck) to obtain a sterile cell-free extract. Protein extracts were prepared for analysis using filter-aided sample preparation[73]. In brief, 50 μg of protein extract was applied to an Amicon Ultra-0.5 mL centrifugal filter (Merck) and washed twice with wash buffer (50 mM ammonium bicarbonate, 8 M urea). Proteins were reduced with 5 mM DTT at 56 °C for 30 min, cysteines alkylated with 25 mM iodoacetamide at 25 °C for 30 min, and excess iodoacetamide quenched by the addition of 5 mM DTT. Samples were resuspended in 50 mM ammonium bicarbonate and digested with proteomics-grade trypsin (Sigma Aldrich) at a ratio of 1:20 at 37 °C for 24 h with shaking (200 rpm). Peptides were isolated following centrifugation, desalted with C18 ZipTips (Merck), and then resuspended in 0.1% formic acid. Approximately 500 ng of desalted

peptides were separated using reversed-phase chromatography on an M-class UPLC system (Waters). Peptides were separated on a NanoEase HSS T3 column (100 Å pore size, 1.8 μm particle size, 300 μm i.d. × 150 mm) (Waters) at a flow rate of 5 μL/min set at 40 °C, with LC conditions as follows: 0–1 min = 3% buffer B (0.1% formic acid in acetonitrile), 1–46 min = 3–45%, 46–52 min = 45–97%, held at 97% buffer B for 4 min followed by re-equilibration for 4 min with buffer A (0.1% formic acid in water). Eluted peptides were directly analysed on a ZenoTof 7600 instrument (SCIEX) using an OptiFlow Micro/MicroCal source. For data-independent acquisition (DIA), an MS TOF scan across 400–1500 $m/z$ was performed (0.1 s). For MS2, variable windows spanning 399.5–750.5 $m/z$ were chosen for fragmentation (0.013 s), with fragment data acquired across 140–1750 $m/z$ with Zeno pulsing on. Dynamic collision energy was used. Sample injection order was randomised. All raw data was acquired using SCIEX OS (v2.1.6).

Peptides were identified using DIA-NN (v1.8)[74], searching against the UniProt reference proteome for Mtb H37Rv (downloaded 13 October 2022, UP000001584). DIA-NN settings were as follows: fixed modification, propionamide (71.037114, C); variable modifications, Deamidation (0.984016, N); Missed cleavages, 1; enzyme (cut), trypsin (K*, R*,! *P). The proteins identified at a 1% false-discovery rate were used to generate ion libraries for analysis of SWATH data. The abundance of peptides and proteins was determined using PeakView (v2.1; SCIEX). Protein abundances were re-calculated by removing all peptide intensities that did not pass a local false-discovery rate cut-off of 1%. Protein abundances were recalculated as the sum of all peptides from that protein and normalised to the total protein abundance in each sample. For statistical comparisons, the PeakView output was reformatted with a custom Python script for use with MSstats (https://github.com/bschulzlab/reformatMS)[75]. Differential protein abundance was compared using a mixed linear model using MSstats (v2.4)[76] in R, using an adjusted significance threshold of $P = 10^{-5}$. Functional enrichment analysis was performed using the online Gene Ontology (GO) resource[77,78] (http://geneontology.org), applying an adjusted significance threshold of $P < 0.05$. Proteins associated with significantly enriched GO terms were visualised using STRING (v11.5)[79].

## Mouse infections

For all animal infections, Mtb strains were synchronised to the mid-log growth phase in 7H9 media. A highly homogenous single-cell inoculum ($10^6$ CFU/mL) was prepared in PBS as described for macrophage infections above. Mice were infected by low-dose aerosol using a whole-animal inhalation exposure system (Glas-Col).

To assess the persistence of the Δ$tcrXY$ mutant, mice were infected with the mutant, parent H37Rv or complemented strain ($tcrXY$-comp). At indicated time-points post-infection, bacterial burden was determined by culturing serial dilutions of organ homogenates on 7H10 agar at 37 °C until colonies were visible (3 weeks).

To assess the persistence of the CRISPRi-$tcrX$ knockdown mutant, mice were infected with a 1:1 mix of CRISPRi-$tcrX$ and its parent H37Rv. To establish a chronic wild-type-like infection, mice were rested for 4 weeks, and then CRISPRi induced in vivo by administration of doxycycline (2000 ppm) in mouse feed (Specialty Feeds, Western Australia). At indicated time-points post-infection, the bacterial burden was determined by culturing serial dilutions of organ homogenates on 7H10 agar with or without kanamycin selection at 37 °C until colonies were visible (3 weeks).

For drug treatment of chronic TB, mice were infected with a 1:1 mix of CRISPRi-$tcrX$ and its parent H37Rv. To establish a chronic wild-type-like infection, mice were rested for 4 weeks, and then CRISPRi was induced in vivo by administration of doxycycline (2000 ppm) in mouse feed (Specialty Feeds, Western Australia). In parallel, a control group received the same base mouse feed composition but lacking doxycycline. All mice were then treated with isoniazid and rifampicin delivered through drinking water ad libitum at 0.1 mg/mL of each,

supplied in light-protected bottles, and replaced every 3–4 days. At indicated time-points post-infection, the bacterial burden was determined by culturing serial dilutions of organ homogenates on 7H10 agar at 37 °C with or without kanamycin selection until colonies were visible (3 weeks).

## Oxidative stress sensitivity profiling

Mtb VICE strains were grown to mid-log growth phase in 7H9 media supplemented with ATc. Bacteria were pre-conditioned to acidified minimal media at pH 5.4 for 24 h, diluted in fresh acidified media to a final $OD_{600}$ of 0.025, and incubated in the presence of 20 or 50 mM hydrogen peroxide for 2 h. ATc was supplemented in all media to maintain CRISPRi activity. The number of viable bacteria in treated and non-treated parallel cultures was determined by CFU enumeration of serially diluted bacteria grown on 7H10 agar plates at 37 °C until colonies were visible (3 weeks).

## Measurement of endogenous ROS

Mtb strains were grown in 7H9 complete media, harvested by centrifugation, and then resuspended in minimal media (pH 7 or pH 5.4) at a final $OD_{600}$ of 0.1. VICE cultures were pre-depleted with ATc for 3 days prior to the assay, and ATc was maintained in the media for the duration of the assay. After 3 days of culture, bacterial strains were again normalised to $OD_{600}$ 0.1 to account for different growth rates in the tested media and then stained with CellROX Deep Red (5 μM) for 2 h at 37 °C. Bacteria were harvested by centrifugation, resuspended in 4% paraformaldehyde in PBS and fixed for 30 min at room temperature. Fixed cells were pelleted by centrifugation, resuspended in 1 mL of PBS, and then analysed by flow cytometry.

## Reporting summary

Further information on research design is available in the Nature Portfolio Reporting Summary linked to this article.

# Data availability

The genome assemblies and sequencing datasets supporting the conclusions from this article have been deposited to the SRA under the BioProject accession PRJNA952396 and the SRA accessions SRR24208443-SRR24208453 and SRR24187079-SRR24187080. The mass spectrometry proteomics data presented in this study have been deposited to the ProteomeXchange Consortium (http://proteomecentral.proteomexchange.org) via the PRIDE partner repository[80] with the dataset identifier PXD041697. Source data are provided with this publication. Source data are provided with this paper.

# Code availability

The code bases that support the findings of this study are available on a GitHub repository: (https://github.com/FordeGenomics/TcrXY_Code). All codes and scripts were uploaded to a TcrXY Code_repository, with the identifier https://doi.org/10.5281/zenodo.10157743[81].

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

## Acknowledgements

We acknowledge the scientific and technical assistance of the School of Chemistry and Molecular Biosciences, the Australia Infectious Disease Research Centre, and the Australian Centre for Ecogenomics. We thank Dr. Amanda Nouwens and Peter Josh, of the SCMB Mass Spectrometry Facility, for their technical support in carrying out the SWATH proteomics. We thank Thom Cuddihy (Forde Lab UQCCR) for providing valuable technical support toward the bioinformatic analysis which included the development of necessary software. Dr. Steven Mason, of the SCMB Microscopy Unit, provided technical assistance for confocal microscopy image quantification analysis, and protocol optimisation. Dr. Igor Makunin, from the Queensland Cyber Infrastructure Foundation, provided technical assistance for RNA sequencing analysis. This work was supported by a Project grant (no. GNT1164198) from the Australian National Health and Medical Research Council (NHMRC), and funding from the Australian Research Council (LE220100068). M. Stupar and C.J. De Voss are supported by an Australian Government Research Training Programme Scholarship. The funders had no role in study design, data collection and interpretation, or the decision to submit the work for publication.

## Author contributions

N.P. West and M. Stupar conceived the study and designed experiments; M. Stupar, L. Tan, E.D. Kerr, B.M. Forde, B.L. Schulz and N.P. West

analysed the data; M. Stupar, L. Tan and C.J. De Voss performed experiments; M. Stupar and N.P. West wrote the manuscript with input from the other authors.

## Competing interests

The authors declare no competing interests.
