## [Peer Review File · Nature Communications]

REVIEWER COMMENTS

Reviewer #1 (Remarks to the Author):

Summary: Overall, this is a well-written manuscript by Stupar et al. detailing a rigorous characterization of the neglected TcrXY two-component regulatory system (2CS) in *Mycobacterium tuberculosis*. It addresses an important knowledge gap in our understanding of Mtb stress responses and host adaptation by defining the role and the regulon of the previously uncharacterized TcrXY. The authors convincingly demonstrated TcrXY is auto-regulated, is induced under acidic pH conditions and within macrophages, and is required for survival and persistence in macrophage and mouse infection models. A fairly large potential regulon was identified by RNAseq analyses, with follow-up investigation of two TcrXY-regulated genes, Rv3706c and Rv3705A, indicating a role in mitigating redox stress encountered under acidic pH conditions. Demonstrating the feasibility of CRISPRi-mediated gene silencing in vivo in Mtb, to my knowledge for the first time, Stupar et al. also showed that disruption of TcrXY signaling in vivo rendered Mtb more susceptible to known TB drugs. These findings support the consideration of TcrXY as a possible adjunctive drug target. However, there were some concerns about methodology used and potential impacts on the data. Specific comments/critiques of some technical details, data interpretation and requested clarifications are described below.

- Results Line 89-107: A dual fluorescent reporter strain is described for monitoring activation of the tcrXY promoter and the legend of Figure 1 (line 138) says that mCherry/GFP fluorescence was measured. However, the data shown in Figure 1 is not ratiometric but looks like mCherry raw fluorescence signal. This needs to be clarified (how was GFP fluorescence used in calculation of mCherry signal shown on graph). The use of “constitutive” GFP as endogenous control is important to account for global changes in protein expression levels likely under stress conditions.
- Line 92: The use of the rpoB promoter for a constitutive control is problematic given that it is known to be repressed under many different stress conditions that limit growth (see Boshoff et al (2004). The transcriptional responses of *Mycobacterium tuberculosis* to inhibitors of metabolism. *J Biol Chem.*). If using as a reference to gauge induction of mCherry, then constitutive behavior of PrpoB should be demonstrated under the conditions used (e.g. acid stress).
- Line 104-106: It is stated that activation of BMDMs increases phagosome acidification, and increased tcrXY transcription was interpreted as proof that pH was the signal cue in the phagosome. Many other parameters will be altered by activation (e.g. ROS, RNI) and no studies were conducted to specifically link phagosome induction to phagosome acidification (e.g. loss of induction of acidification blocked or buffered). It would be good to temper the interpretation of this data accordingly.
- In Figure 1g (and line 118), it seems that the TcrX D54A mutation had a lesser impact on signaling than TcrY H256Q mutation. If both mutations block Phosphotransfer, this observation seems unexpected. Could the authors provide any comment on this point?

- In Figure 3 and corresponding text, TcrXY is described as predominantly a repressor, yet three genes that appeared to be activated by TcrXY (minority geneset) were selected for followup. Providing a rationale for this choice would be helpful.
- Line 221: The loss of Rv3706c induction by acid stress in TcrX/Y phosphorylation mutant suggests that phosphorylation of the response regulator is crucial for promoter interactions. However, EMSA studies (Fig 3g) seem to show that unphosphorylated TcrX was able to bind target promoters, yet this observation was not discussed. Is phosphorylation required or not? Was phosphorylation (e.g. with acetyl phosphate) of TcrX attempted for EMSA studies?
- (minor): Figure 5C – The labeling of the Y-axis as Lung survival (%) is confusing. How is this data different than the CFU/lung data shown in Fig 5A. One is CFU enumeration in lung after CRISPRi induction and the other is CFU after drug treatment.
- Discussion Line 429: There is fairly long discussion about SPRP family of proteins from eukaryotes. The connection to tarA/tarB is not clear at all. What is homology level with SPRP proteins? Are tarA/B even protein rich? The discussion would benefit from clarifying the relevance of these comments.
- (minor): Line 447-448: This explanation for why CRISPRi knockdowns were used vs k/o would have been helpful to include earlier in the results section to make clear the rationale for those experiments. The reader may be left wondering (as I was) what was the benefit of introducing a second more complicated, less effective (partial knockdown) strategy.
- Was any attempt made to identify a TcrX binding motif? This would aid in prediction of members of the regulon controlled directly by TcrX vs. indirectly.
- (minor) Line 518 (and 554, and throughout): Suggest the use of the term CRISPRi hypomorphs rather than mutants since no gene is actually mutated.
- (minor) Line 554: Should read “Mtb CRISPRi hypomorphs were pre-induced with ATc 3 days prior to infection”. Technically, as is this sentence says “prior to infection with ATc” and of course ATc is not infectious :)
- Line 604: Similar to concern raised above in regards to GFP controlled by rpoB promoter, the use of rpoB as a constitutive housekeeping gene raising significant concerns about mRNA induction studies. It is suspected that repression of rpoB under acid stress or intramacrophage conditions will falsely elevate the apparent induction of tcrXY and downstream genes.
- It is a bit unclear how doxy was administered to mice. In Supplemental Fig 9 legend (line 111) it says via drinking water whereas on line 377 in main text (Fig 5 legend), line 710, and line 717 it says doxy was given in mouse chow. Please clarify if different routes were used for different experiments.

Reviewer #2 (Remarks to the Author):

Stupar et al. report the analysis of an autoregulated acid-sensing two-component system that is required for persistence of *M. tuberculosis* (Mtb) during chronic mouse infection. They identified the TcrXY regulon and provide evidence that the two of the TcrXY induced genes (*rv3705A*, *rv3706c*) are important for alleviating oxidative stress. Finally, they report that knockdown of *tcrXY* increased the activity of isoniazid and rifampin during chronic mouse infection.

Overall, this is an interesting body of work that provides new insight into the mechanism by which Mtb withstands acidic environments. There are however several experiments that lack the appropriate controls and some of the conclusions are not rigorously supported by the presented data. These issues need to be addressed.

Major comments:

1. The authors propose that TcrXY is a novel target for enhanced TB therapy, which would be exciting. However, the data supporting this conclusion are difficult to judge and interpret. There are 3 issues:

(a) The kill kinetics of wt Mtb and the CRISPRi mutant in the absence of doxycycline mimic the kill kinetics of the CRISPRi-TcrXY mutant in the presence of doxy, i.e they resemble the kill kinetics with doxy in the left panel. This suggests that the knockdown of *trcXY* does not affect the impact of isoniazid and rifampicin and raises the concern that doxy affects the impact of INH and RIF (drug-drug interactions that protect wt Mtb from RIF and INH). It is unclear why the authors did not conduct this experiment with the deletion mutant; the *trcXY* knockout ($\Delta trcXY$) is specifically attenuated during chronic infection (i.e. when the authors induced *trcXY* silencing) and thus ideal for addressing the question if killing by INH and RIF is accelerated in the absence of TcrXY compared to wt Mtb.

(b) Lack of complementation. The authors did not include a CRISPRi resistant strain, and it is possible that off-target effects contribute to the impact seen with doxy.

(c) Co-infection amplifies in vivo attenuation. Using $\Delta trcXY$ will also not require a co-infection (I assume the authors conducted co-infections to prevent to emergence of CRISPRi escape mutants), which enhances in vivo survival defects due to competition. The comparison of wt and $\Delta trcXY$ with drug treatment in vivo is a critical experiment to support the authors conclusions.

(d) The interpretation that “Rv3706c and Rv3705A mitigate redox stress to permit Mtb persistence” needs to be better supported. The impact of *rv3706c*/*Rv3705A* depletion on endogenous ROS production and resistance against H₂O₂ is minor and no attempts were made to complement these phenotypes. This is critical, because there is no control for the impact of CAS9 expression (the control is just an empty vector). The authors must complement their mutants with CRISPRi resistant alleles to demonstrate that the reported phenotypes are indeed due to knockdown of the targeted genes. This is

also important because it addresses the issue of polarity; it is possible that knockdown of rv3705A also leads to knockdown of rv3706.

(e) The authors nicely show that tcrXY is not hypersusceptible to growth inhibition by INH and RIF in culture. However, the relevant question is whether the mutant is hypersusceptible to killing by these antibiotics. The authors should determine the INH + RIF induced in vitro kill kinetics of Δ tcrXY compared to wt and complemented strains.

Minor comments

1. Figure 1g needs statistical analysis. It seems as if the loss of a phosphorylation site has a larger impact on TcrX than TcrY. How do the authors explain this?
2. The authors show impact of mutating the phosphorylation sites on the ratio of trcX mRNA at acidic and neutral pH. They need to also show if the basal mRNA abundance at pH 7 is affected by the mutations.
3. Figure S6 needs statistical analysis.
4. Do the promoter regions of rv3705A, rv3706 and rv0666 contain a TrcX binding motif?

Reviewer #4 (Remarks to the Author):

Summary

This is a comprehensive study of the two-component system (TCS), *tcrXY*, with many corroborating lines of evidence. Though this TCS can be considered 'understudied', some of the results presented here corroborate the existing literature, confirming induction of the system at low pH, autophosphorylation of *tcrY* by *tcrX* and direct binding of TcrX to the *tcrXY* promoter. The authors further expand knowledge of the system by using reporters and microscopy to demonstrate that induction of the system occurs in acidified compartments in BMDMs. A knock-out system using a hyg cassette demonstrates that inactivating the *tcrXY* system abrogates pH responsiveness and that "complementing" with phosphorylation-deficient *tcrXY* alleles does not restore phenotype (as expected).

The authors go on to perform RNA-seq to show changes in gene expression between the *tcrXY::hyg* mutant and wild-type H37Rv and choose two of the differentially expressed genes (Rv3706c and Rv3705A) for further study. A mCherry transcriptional reporter of one of the targets, Rv3706c, shows that transcription of Rv3706c is positively linked to functional TcrXY and low pH in culture and co-located in the acidic compartment of macrophages. EMSAs with purified TcrX confirm direct interaction between the TcrX and the Rv3706c promoter. CRISPRi mutant strains of the two target genes were made that appear to show regulation by *tcrXY*. The proteomes of each of the silenced strains were evaluated with MS-SWATH and there is evidence for a shift in the proteome which was correlated between the two silenced strains. The CRISPRi strains were tested for their ability to manage oxidative stress and demonstrated diminished ability to prevent accumulation of endogenous ROS. Using an *in vivo* mouse model with a third CRISPRi strain to silence *tcrXY*, they authors demonstrate a role for the system in persistence and also provide evidence for the suitability of inhibition of the system to augment clearance by rifampicin and isoniazid. The noteworthy results in the publication are regarding the role of these two regulated genes in ROS protection as Rv3706c is a gene up-regulated in several models of infection as well as human granulomas (doi: [10.3389/fmicb.2010.00121](https://doi.org/10.3389/fmicb.2010.00121)). This will pique interest in the field as the function of this gene has not been predicted by sequence similarity.

Validity: how robust are data and conclusions. Any flaws that prohibit publication?

Though there is an extensive amount of experimentation performed, not all the assertions that the authors have made are supported by the evidence presented. The claim that the authors have

identified a TcrXY-regulated regulon composed of 117 genes that is "required for bacterial persistence" is unsupported. Firstly, a 1.4-fold change (with an p-adj threshold of 0.05) is a rather low threshold for identifying the differential expression of genes under different environmental conditions, especially in stress conditions that result in a lot of transcriptional noise. A fold-change of at least 2X would be more reliable and informative. Applying a cut-off of 2X would give a much smaller number of genes (less than 10) regulated. Given that there are no experiments that further confirm direct regulation (see below) then the claim that TcrXY has a 117 gene regulon is unsubstantiated.

Furthermore, the methodology for quantifying gene expression is inappropriate. The RNA-seq analysis pipeline referred to in the Methods (<https://usegalaxy.org.au/u/mstup1/w/rnaseq-analysis>) indicates that the strandedness parameter used with featureCounts was set to 'unstranded'. Therefore, all reads mapping antisense to coding genes will be attributed to the coding gene. Antisense expression increases in Mtb in response to stress conditions and may account for some of the global difference between pH conditions in knockout setting. This should be repeated with the correct parameter setting.

The authors also do not make a distinction between direct and indirect regulation of the members of the proposed regulon. Apart from Rv3706c, no direct binding interactions between TcrX and the target genes were explored and despite the recognition of a known DNA-binding motif (the 7-bp 'mirror' repeat), there is nothing reported about whether this motif exists in the promoters of any of the proposed gene targets. It seems highly speculative to claim to have discovered a large gene regulon given these omissions.

Significance

This publication claims to be characterising an 'understudied' and 'neglected' 2CS. Though this work does advance the understanding of this system and its possible applications to treatment of tuberculosis there is relevant research prior to this that the authors do not sufficiently credit. For example, there is biochemical work that quantifies binding interactions between *tcrX* and *tcrY*, verifies autophosphorylation of *tcrY* and performs phosphorylation assays of *tcrX* and *tcrY* (Bhattacharya et al, 2010 and Bhattacharya et al, 2011). The same group also identified and tested the putative DNA-binding motif of TcrX and created multiple sequence alignments of the system across mycobacteria. Likewise, a paper in 2004 demonstrated that *tcrXY* was expressed in late phase (mature) human macrophages (Haydel and Clark-Curtiss, 2004). Repeating this work is useful but should not be presented as novel. The identification of a direct target of *tcrXY*, Rv3706c (and potentially Rv3705A, as

well), and its link to the regulation of redox stress is a significant and novel discovery, and there are multiple lines of evidence to support these conclusions.

Data and methodology: assessment of validity of approach, including all data even supplementary materials

The data are publicly available, including RNA-seq and proteomic analysis and the Supplementary materials include spreadsheets of the differentially expressed genes and proteins.

The RNA-seq analysis has an issue with quantification of the reads and the threshold for differential expression addressed earlier in 'Validity'.

The qRT-PCR experiment in Figure 1g compares transcription of *tcrXY* in the knockout with additional alleles of WT and phosphorylation mutants and concludes that phosphorylation of both TcrX and TcrY are required for transcription of *tcrXY*. However, the allele-complement strain was not directly compared to the WT.

The EMSA showing binding of purified TcrX to the promoter of Rv3706c (Figure 3g) does not indicate the DNA:protein ratio in the figure, legend or in the Methods. A very high ratio of DNA to protein can cause non-specific binding.

There is a lack of biological replication (only technical replication) in Supplementary Figure 1: *tcrXY* mRNA is induced at acidic pH. However, the data is in the supplementary and the experiment was largely designed to corroborate several lines of published evidence.

Statistical tests are inconsistently applied. For example, qPCR confirmation of 3 of the differentially expressed genes (Supp figure 6) does not include a test of statistical significance, though it has 4 replicates. Are the changes in expression levels for Rv3705A and Rv0666, statistically significant? Other data, in particular, mCherry quantification in Figures 1f and 3d, and in Figure 4a: CFU count per well over time, include only two replicates but do apply a t-test to determine statistical significance. In other tests that have only two biological replicates, such as the qRT-PCR experiments in Supp Figure 8 and Supp Figure 9b, no statistical tests of significance are made. Statistics such as p-value and standard deviation have no meaning with fewer than 3 data points.

Suggested Improvements

Address the issues raised in previous sections:

- re-run feature Counts with stranded library settings and repeat the differential expression with more accurate quantification
- raising threshold for fold-change to at least 2X would be more robust and may be more informative in respect to the functional categorisation of the genes in the regulon
- apply statistical tests consistently
- examine promoters of proposed *tcxY* gene targets for binding motif including any others found to be differentially expressed.
- Direct binding studies of Rv3705A: is it regulated by its own promoter and does this promoter bind TcrXY directly (EMSA)?

Clarity and Context

- The allelic recombination mutant constructed strictly speaking isn't a deletion mutant – therefore should not be annotated as $\Delta tcxY$.
- The y-axes for the plots in Figure 4f: CRISPRi-mediated silencing of Rv3706c or Rv3705A enhances Mtb sensitivity to oxidative stress at acidic pH are confusing. The scale should be consistent between plots.
- In the figure legend for Figure 3, there appears to be a typo with respect to the Rv3706c reporter; also found in text, line 212. It is assumed the H37Rv control strain also included the same reporter (PRv3706c) as the *tcxY::hyg* strain?
- make available all R-scripts (such as those using MS-stats) for the statistical analysis of MS-SWATH proteomic data.

To summarise, while this study does contain a lot of experimental data, additional evidence for the claim that *tcxY* controls a 117 gene regulon is the major flaw in this paper. As such, this detracts from the impact.

Step-by-Step Responses to Reviewer Comments Re:

Stupar et al. TcrXY is an acid-sensing two-component transcriptional regulator of Mycobacterium tuberculosis required for persistent infection.

Reviewer 1

Line 89-107. Agree. Fig 1a and 1b have been redrawn to show both mCherry and GFP expression.

Line 92. Agree. The level of background RpoB expression should be reported and is now included in Figs 1a and 1b.

Line 104-106. Agree. We have tempered the language relating to BMDM acidification as the only cause of TcrXY induction. Text has been added to Line 107 indicating a suggested relationship only - “suggesting a causal, intracellular link”

Fig 1g (and line 118). Agree. The outcome (Fig 1g) does appear counterintuitive; however, the fact remains that transcription level are differentially altered due to the influence of phosphorylation. These results suggest that the system is clearly intricate and not fully understood, with the possibility of other factors affecting their expression. We added this text to clarify our intended message at Line 122.

Transcriptomic analysis and Fig 3. Agree. We have more appropriately designated the regulator to now read “predominantly a negative regulator” as opposed to “largely a transcriptional repressor” (Line 193). The section has also been reorganized to remove the perceived conclusion that gene repression is the greatest consequence of TcrXY activity. We believe the section reads more easily now.

Line 221. We thank the reviewer for this insightful comment. We agree more was required and we performed additional experimental work. We performed in vitro phosphorylation of TcrX with acetyl phosphate which produced outstanding, unequivocal evidence of the impact that phosphorylation of TcrX has to its own activation and that of *Rv3706c* and *Rv3705A* (*tarA* and *tarB* respectively) and their designation as regulon members. These results are now included in Supplementary Fig 4. These studies have also allowed us to correct the currently held view that TcrX – DNA interaction is independent of phosphorylation state (Bhattacharya *et al.*, 2011, *Biochem Biophys Res Commun* **415**, 17-23).

- Note on Fig 3. Due to our additional EMSA work to address reviewer question, we have re-drafted Fig 3. The EMSA result has been removed and included in a larger figure dedicated to TcrX-promoter binding and phosphorylation (via EMSA). We have added this figure to Supplementary figures as S.Fig4. Fig 3. Is now dedicated to transcriptomics outcomes.

Minor; Fig 5c. Agree. The axes are confusing in that they do not identify the group as antibiotic (RIF, INH) treated. The axes have been modified to include RIF and INH treatment on x-axis, which removes any confusion with that shown in Fig 5a. The explanation of Fig 5c remains in the figure legend at Line 384.

Line 429. In our discussion of TarA/B and SPRP proteins, the link has been made stronger by reminding readers that the Tar proteins have been annotated as conserved proline-rich proteins, with references added (Line 214, Line 430). We have reworded the opening sentence to the discussion paragraph (Line 430), with the following text - “We provide here the first description of two novel TcrXY regulon members, *Rv3706c* and *Rv3705A*, both annotated as putative

proline-rich proteins”. We believe the remaining discussion is sound and are only suggested mechanisms for impact of these proteins. We have therefore not made further changes.

Minor; Line 447-448. Agree. However, there are many strong reasons as to why CRISPRi strains are preferred over gene deleted strains and listing all at any one point in the paper is distracting. As a result we have chosen to introduce the relevance of the approach at different points. To that end, we have added a sentence at Line 268, in line with reviewer’s suggestion. The sentence reads, “Inducible CRISPRi developed by Rock and colleagues for use with Mtb is a powerful tool for investigating Mtb biology and pathogenesis, allowing for the rapid interrogation of newly identified genes of interest, without the time-consuming need for gene deleted and complemented strain generation.”

TcrX binding motif. Agree. Yes, since receiving reviewer feedback (Reviewer 1 and 3) we agree that this would strengthen any claims of regulon membership. We have written new computational script for this purpose and ran a screen, but were unable to identify a consensus sequence with high confidence. The relationship between tcrX-promoter interactions appear intricate and require further investigation, which is outside the scope of the present study.

Minor; use of the term hypomorph suggested. Line 518 and elsewhere. Agree. The term mutant is incorrectly used here. Unfortunately we do not feel the term “hypomorph” is the most appropriate term in this instance. We now use the acronym, VICE, for Variant by Inducible CRISPRi Expression strains, or simply “CRISPRi strains” at times. Line 272-273, 297, 321-342, 349, 571, 656 and 731.

Minor; Line 554. Agree. Text has been corrected. (now Line 571)

Line 604. As stated above, we have tested the variance in *rpoB* promotor activity and found to be stable. This is demonstrated in Figure 1a and b. No changes made to text.

Doxy administration. Yes doxycycline was administered to mice via different mechanism depending on experiment. For earlier experiments, we used doxycycline in drinking water, as described in Gandotra et al., 2007. Nat Med. 13:12. Pg1515. We changed to delivering Doxy in chow for all subsequent experiments, as described in Tiwari et al., 2018. Sci. Transl. Med. 10. Pg1-10. We have found no differences in the effect of the drug via either route, however for practical reasons, chow delivery is our preferred method. Method of delivery is mentioned where relevant throughout manuscript.

Reviewer 2 – Major Comments

1a. We agree that it is very important to know whether drug-drug interactions are apparent, and whether the mutation (CRISPR induced or otherwise) have altered susceptibility to RIF and INH. This precise question was tested *in vitro* and reported in Supplementary Figure 10. As seen in Supp Fig 10, MIC curves for all drug combinations are identical.

For Fig 5c, the result is therefore very meaningful, with TcrX suppression leading to a significantly additive effect on *in vivo* survival, especially when one considers that this is a co-infection, with both strains infecting the same animal (WT and CRISPR-tcrX strain). We believe this is the most stringent way of demonstrating the significant impact of the additive value of TcrXY inhibition (doxy induced) to that observed with RIF + INH only. It may be that the reviewer has not realised that figure is reporting a co-infection model, with all bacteria in the

mouse being subject to the same treatment. Those having TcrXY forced repression display rapid clearance (kill kinetics). The use of CRISPRi to demonstrate the time sensitive gene knock-down, i.e. during chronic infection and not before, was in fact the point of the experiment. This is not possible with gene knockouts. We can mimic a clinical treatment scenario and demonstrate the potential utility of a specific TcrXY inhibitor (should one be developed). It is also essential to appreciate that this is a demonstration of how this technology can be used to identify *in vivo* drug targets more broadly.

1b. Agree. The lack of a CRISPRi-resistant complement is warranted. We have since constructed these strains and tested these in an intramacrophage persistence model, as shown in Fig 4a. Our repeat of this experiment has shown that introduction of a CRISPR-resistant allele corrected any survival deficit, thus proving a lack of off target effects of the sgRNA when induced. We have updated Fig 4a with this new data. We have also added the method of complement (CRISPRi resistant) construction to the Method at Line 533-538.

1c. We respectfully disagree. We do not believe that enhancement of *in vivo* attenuation due to competition is a flawed concept. We conducted these co-infections for reasons stated above, but also to calculate a competitive index, which was reported in Fig5b, d (now Fig 6b and d). This is a very well accepted method of defining the impact of a mutation, or in this case gene silencing event, to obtain direct quantitative comparisons. Additionally, we do not believe that the data generated from a repeat of this work with a KO strain, warrants the exceptionally long experiment durations required (18-22 weeks), nor do we believe it would be an ethical use of animals to do so for the suggested outcome.

1d. Agree. Complementation with a CRISPR-resistant allele is important. As with response to point 1b above, we have also completed this additional work and it is now included in Fig 4f. It is worth noting however that in contrast to the statement “there is no control for CAS9 expression (the control is just an empty vector)”, our control vectors do express CAS9 constitutively - it is only the sgRNA which is not encoded.

To address reviewer comments relating to a need to better support the notion that *Rv3706c* and *Rv3705A* mitigate redox stress to permit *Mtb* persistence, we have undertaken additional experimental work. We now include proteomic results detailing the impact of *Rv3706c* silencing at acidic pH. This section adds significantly to the justification of the statements. Proteomic changes are strongly in favour of the conclusions. The data has been added to Fig 4, along with the relevant principle component analysis of the experiment.

1e. Agree. We have completed additional experimental work to assess the kill kinetics of the *trcXY* knock out mutant and complement. The results have been included with Supplementary Figure 10, demonstrating that the mutant is not hypersusceptible to RIF and INH as reported by direct bacterial viability.

Minor Comments

1. See response to Reviewer 1 above (Figure 1g) and added text at Line 122. Statistical analysis has been added to Figure 1g.
2. Basal mRNA levels of *trcX* were stable across all experimental groups (WT complemented and phosphor-mutants) at pH7. In fact, the mean Ct values of these groups varied by less than 1 cycle (SD 0.34) in 5 biological replicate experiments of at least triplicate assays.
3. Stats for Supp Fig 6 needed – Stats added to figure, now Supp Figure 7.

4. The promoters of Rv3705A, Rv3706c and Rv0666 do not include the identical TcrX binding site as it appears within the promoter region of *tcrX*. Please see comments to Reviewer 1 at section “TcrX Binding Motif”.

Reviewer 3. General comments

Validity

Comments of unsupported claims. We agree with the reviewer that more careful language could be used, and that many of our findings (gene lists) can not be simply proven as uniquely fitting into the TcrXY regulon collection. We have considered these comments carefully and are happy to report that we have made several major corrections (outlined below – suggested improvements section)

Significance

“There is relevant research prior to this that the authors do not sufficiently credit”. We disagree with this statement. We have referenced the said article pertaining to identification of TcrX binding motif (Line 131), and have added another citation at Line 135. We have not simply repeated their results. This group did not identify pH as a critical trigger for phospho-communication and did not perform assays under differing pH conditions. In response to reviewer suggestions, we have now more thoroughly assessed the impact of component phosphorylation and are now reporting contrasting results to those reported by Bhattacharya, i.e. phosphorylation of TcrX directly impacts the affinity for binding (Line 129), a novel finding.

Data and methodology.

RNAseq analysis comment. Addressed above in “validity”

qRT-PCR comment. Figure 1g has been modified to include WT to allow for a direct comparison.

DNA:Protein ratio not given in EMSA. 10nM of DNA was used in all EMSAs. EMSA methods have been removed from the manuscript and added to Supplementary file. This was to accompany the new EMSA figure, Supplementary Figure 8 and the adjusted EMSA figure, S4. DNA concentration used now added to both figure legends and methods (Line 246 – Supplementary data file)

Suggested improvements

1. We have reanalysed the RNAseq data, with the suggested settings implemented, i.e. the strandedness parameter used with featureCounts (Galaxy) was set to stranded. This has had the effect of strengthening the magnitude of differential gene expression, enabling more accurate quantification.
2. We subsequently also increased the fold change cut-off to 2X, which has reduced the number of reported genes to 70. Results shown in updated Figure 3a and b. Results (Line 187-201) and discussion adjusted accordingly.
3. Statistical tests. Outcomes of all statistical test requested now shown.
4. In relation to the possibility of binding motifs, please see above response to Reviewer 2, minor comment 4.
5. We have extended our analysis of TcrX-promoter binding via EMSA to include Rv3705A and demonstrates these two genes are independently TcrX activated. Furthermore, we have also demonstrated that phosphorylation of TcrX enhances binding affinity. These novel findings have been included in Supplementary Fig 8 and also stated in text at Line 219-225.

-
6. “The allelic recombination mutant constructed strictly speaking isn’t a deletion mutant”. We respectfully disagree. Out knockouts are produced via allelic exchange, which replaces the entire target gene with a gene of antibiotic resistance. This is widely accepted as a deletion mutant.
 7. “Y-axes for plots in Fig 4f – scale should be consistent”. We respectfully disagree. The scale of axes need to be different for purposes of illustration. Axes of the same scale would make it impossible to see reported variation between strains.
 8. Rv3706c reporter typo. Agree, and we thank the reviewer for picking this up. The text (line 234) and figure legend have been corrected to indicate that the WT H37Rv strain included the same Rv3706c reporter.
 9. All R-scripts have been made available.

Reviewer 4. No comments.

REVIEWER COMMENTS

Reviewer #1 (Remarks to the Author):

The authors did a very thorough and rigorous job of addressing all of my concerns/comments as well as those of other reviewers in my opinion. The additional experiments conducted to address the effects of phosphorylation on TcrX DNA binding and to include CRISPRi-resistant complementation strains further enhanced the content of the study. Revisions to the text and figures were effective in making data presentation clearer and aligning conclusions and interpretations with the data. The responsiveness of the authors to constructive criticism is much appreciated and resulted in improving an already strong and interesting story.

Reviewer #2 (Remarks to the Author):

Stupar and colleagues strengthened their manuscript with new data and additional analyses. I am favor of publication, except for one issue that is still not appropriately addressed or discussed. Unfortunately, the authors may not have understood my previous criticism. In Fig. 5 C the authors compared killing of WT and the TcrX CRISPRi mutant in mice treated with INH+RIF in the presence (induction of TcrX silencing) and absence of doxycycline. They interpret the difference in kill kinetics in the presence of doxy as demonstration of an additive effect of TcrX depletion and RIF+INH activity. This is a reasonable interpretation. However, it is flawed when taking the kill kinetics in the non-doxy mice into account. As expected, both strains are being killed equally and with the same kinetics and CFU at the end of the experiment, but the kill kinetics and CFU at the different time points mimic those observed with the TcrX mutant in the presence of doxy. I.e. it seems that doxy treatment protected and slowed killing of the WT instead of accelerating treatment impact on the mutant. At a minimum, the authors need to discuss this unexpected result in the manuscript. Ideally, they would repeat the experiment with the TcrX deletion strain which would not require the addition of doxycycline and could conclusively determine the impact of INH+RIF in the presence and absence of TcrX and significantly strengthen the potential impact of their work.

Reviewer #3 (Remarks to the Author):

The authors have successfully addressed the major issues of the original manuscript, including reanalysis of RNA-seq and additional experimental work to support the conclusions that both target genes are

regulated directly by tcrXY. The work is very exciting and of great interest to the Mycobacterial community.

Two issues remain unresolved in the revised manuscript:

1) In respect to our comment:

"despite the recognition of a known DNA-binding motif (the 7-bp 'mirror' repeat), there is nothing reported about whether this motif exists in the promoters of any of the proposed gene targets"

the authors have written in their rebuttal:

"The promoters of Rv3705A, Rv3706c and Rv0666 do not include the identical TcrX binding site as it appears within the promoter region of tcrX. Please see comments to Reviewer 1 at section "TcrX Binding Motif"."

And:

"TcrX binding motif. Agree. Yes, since receiving reviewer feedback (Reviewer 1 and 3) we agree that this would strengthen any claims of regulon membership. We have written new computational script for this purpose and ran a screen, but were unable to identify a consensus sequence with high confidence. The relationship between tcrX-promoter interactions appear intricate and require further investigation, which is outside the scope of the present study"

We agree with the authors but this information (along with the scripts) needs to be present in the manuscript. As two of the three reviewers were curious about the same point, it would seem to us that this information would likely be of interest to many readers and should be included in the paper. The fact that the downstream targets of tcrXY, including those that have been shown to directly bind TcrX (EMSA), do not use the identified autoregulatory binding motif is very interesting to anyone studying two-component systems in Mtb (as this is different from other well-studied TCS like PhoPR, for example). The lack of a consensus binding site in the other members of the regulon is likewise relevant as it may indicate more complex interactions, as the authors comment in the rebuttal. Furthermore, the computational script used to search for possible binding motifs should be made available with the other scripts used in the study.

2) The authors mention that all R scripts are available as a 'supplemental document' but there is no mention of this document in the manuscript, and we were unable to locate it in the supplemental

materials. Ideally, scripts should be made available on a Github repository so they can be easily downloaded. We would like to see the scripts using 'MS-Stats' and for searching the putative regulon for motifs made available. Obviously, if any web services like 'Meme' (<https://meme-suite.org/meme/tools/meme>) or some other software tool was used, this should be referenced.

Reviewer #4 (Remarks to the Author):

Step-by-Step Responses to Reviewer Comments_B Re:

Stupar et al. TcrXY is an acid-sensing two-component transcriptional regulator of Mycobacterium tuberculosis required for persistent infection.

Reviewer 1: We thank the reviewer for the kind comments.

Reviewer 2: We appreciate that the reviewer has considered the data critically and made this interesting observation. After reflecting carefully on this, apart from intra-experimental variation, we strongly believe the most likely reason is related to the palatability of doxycycline. It is known that the bitterness of doxycycline, when added to drinking water, needs to be offset by the addition of sucrose to encourage its consumption, and this is our experience also. This is one reason that we now use doxycycline in mouse chow, as we did for this experiment. However, these results are perhaps suggesting that the consumption of doxy-containing chow is also reduced and is in turn affecting the amount of water consumed. This is of course relevant here because the RIF and INH are administered in drinking water. The amount of feed and water consumed by mice is correlated (Bachmanov A., et al., *Behav Genet.* 2002; 32(6): 435–443) and the intake of chow is thought to stimulate water intake due to multiple mechanisms for stimulation of thirst (Kraly FS. *Psychol Rev* 1984;91:478–490). Therefore, a reduced chow intake could result in a reduced RIF/INH dosage and affect the kill kinetics of Mtb in these mice, causing it to look as if the WT was “protected” with its killing “slowed” (Fig6c).

Critically, as this experiment was conducted as a co-infection, all bacteria are subjected to the same selective pressures, i.e. WT *Mtb* and the *TcrX*-CRISPR KD strain alike, thus validating our result.

However, we agree this is a finding worth considering by readers when planning and analysing a doxy-induced CRISPRi knock down study in mice and have therefore added the following text to the manuscript to highlight.

Line 469: A slight reduction in clearance rates of Mtb was observed in RIF/INH treated mice when fed doxycycline-containing chow (Fig 6c). It is possible that doxycycline reduces the palatability of the chow, as it does when added to drinking water, which requires the addition of sucrose to encourage consumption. A reduction in food intake may lead to a reduced intake of drinking water and hence antibiotic dosage. While this effect was only mild, and does not impact the interpretation of outcomes reported, it should be considered when conducting doxycycline-induced CRISPRi gene silencing in vivo.

Reviewer 3.1: Agree. The lack of an identifiable binding motif is of interest to readers. We have added the following passage to the manuscript. We have also added the script used in the genome-wide search to the script repository, as recorded in “Code Availability”.

Line 225: Interestingly, the promoter regions of these genes do not contain the TcrX binding motif, as is present in the tcrXY promoter. Consequently, we devised and conducted a bespoke genome-wide in silico screen, without revealing any obvious binding motif consensus. While more work is required to elucidate promoter-response regulator interactions, it appears that the mechanism employed by TcrX may be atypical in relation to its downstream regulon genes.

Reviewer 3.2: Agree. We apologise that the scripts used were not available at time of review. They were uploaded to the submission portal as “Supplementary Script” however it seems that they did not make it through for Reviewer consideration. We have now made all scripts

Step-by-Step Responses to Reviewer Comments_B Re:

Stupar et al. TcrXY is an acid-sensing two-component transcriptional regulator of Mycobacterium tuberculosis required for persistent infection.

available on a Github repository, including that for MSstats and our motif filter. These are available through a repository link and a DOI, now added to “Code Availability”.

Reviewer 4: We thank this ECR reviewer for their contributions to the assessments of our manuscript.